



A question of time and space: A model approach to the synchronicity of gypsum and halite during the Messinian Salinity Crisis

Ronja M. Ebner[1], Paul Th. Meijer[1]

[1]Department of Earthsciences, Utrecht University, Utrecht, 3584 CB, The Netherlands

*Correspondence to*: Ronja M. Ebner (r.m.ebner@uu.nl)

**Abstract.** Saltgiants, although well studies, still offer some unsolved questions. One example is the Messinian Saltgiant which formed during the Messinian Salinity Crisis (MSC, 5.97 to 5.33 Ma) in the Mediterranean Sea. While a common assumption is that gypsum precipitated in the marginal parts of the basin before halite formed in the deeper part of the basin, this could be not yet been confirmed. Indeed, it has also been suggested that, while

the primary lower gypsum was forming, the deep basins was already accumulating halite. In this study we use box modeling to investigate the distribution of halite and gypsum deposits for different configurations. Due to a dimensionless description of basin restriction, our results can be transferred to other basins. With this approach we find that under the right conditions all configurations lead to a simultaneous but spatially separated precipitation of gypsum and halite. They would, however, not lead to the spatial pattern that is observed in the

Mediterranean, i.e. halite deposition in the deep basins while gypsum is deposited in the margins. Based on those results we propose a timeline for a salinifying basin. For an average salinity above gypsum but below halite saturation, halite is first formed in a sufficiently restricted margin, and only once the average salinity approaches the one of halite saturation can it also form in open areas of the basin due to horizontal salinity gradients. Once the whole basin has reached halite saturation, gypsum only forms in margins with a positive local freshwater

budget. Such a mechanism would produce less than 1m of gypsum within 25kyr. We thus conclude that a simultaneous, yet spatially separated precipitation of gypsum and halite within a one basin is possible, but unlikely to have led to the massive primary lower gypsum outcrops in the Mediterranean, while halite formed in the deeper parts of the same sub-basin.

## 1 Introduction

Although the Mediterranean is known for its equable conditions, this does not apply on geological time scales. In fact, at the end of the Miocene the Mediterranean Sea was in an extreme state compared to today, leading to the youngest known salt giant to form. This event, called the Messinian Salinity Crisis (MSC) (Hsü et al., 1973;


Ryan, 2009), has been the subject of study for more than 50 years and took place in a geologically short timespan (5.97 to 5.33 Ma, (Roveri et al., 2008) ). This has been determined by using astronomical tuning on the onshore MSC record (Krijgsman et al., 2001) leaving a gap of 600 kyr.

The sedimentary record of the MSC includes gypsum deposits in the marginal parts of the basin and in the deep basin mainly halite, which reaches up to three km in thickness and adds up to $821 \pm 50$ thousand cubic km of late Messinian salt (Haq et al., 2020). In the stratigraphic consensus model, the succession of those evaporites is divided into three stages. In this three-stage model the deposition of the thick halite unit is both preceded and succeeded by a period of dominantly gypsum precipitation (Roveri et al., 2014). In this model the MSC is assumed to start with gypsum precipitation in the marginal basins (stage 1, duration 0.371 Myr) which is then followed by halite precipitation in the deep basin (stage 2, 0.05 Myr). At the end of the crisis (stage 3, 0.22 Myr) the salinity decreased again, and the system experienced another phase of gypsum precipitation (Roveri et al., 2014).

However, the correlation between the various sedimentary units is never unambiguous, since we, for example, cannot follow the layers of the so-called *primary lower gypsum* (PLG) from the margins to the deep basin. Whereas the iconic stratigraphic section for the Caltanissetta basin of Sicily shows halite to overlie gypsum (Decima & WEZEL, 1971; Decima & Wezel, 1973) these different units are found (in well core and mine) in places removed by some horizontal distance and their lateral correlation is not observed. That is, we cannot really exclude the possibility that these two units are each other's lateral equivalent. This idea seemed to get reinforced by a recent study in the Levantine basin (Meilijson et al., 2019), but another study by (Manzi et al., 2018) conducted on data in the same area saw the three-stage model confirmed. A more recent study, however, re-opened this question again (Oppo et al., 2023). This goes to show the complexity of the problem. The coevality of the primary lower gypsum and halite would have implications for the duration of precipitation of the latter and thus, would make room for new scenarios.

For these kinds of problems modelling can add valuable insight. They allow us to test interpretations and hypotheses against the principles of physics and explore the behaviour of systems theoretically and in a way that is transferrable to similar systems. As such, it is the objective of this paper to apply a model approach to the question whether it is physically possible that gypsum and halite deposits formed in different depth ranges of the Mediterranean basin at the same time, by some form of salinity gradient. A similar approach was adopted by (Simon & Meijer, 2017) who investigated the spatial distribution of salinity using a box model with prescribed rate of overturning. The latter leaves open the question whether such overturning would actually develop. It was found that a significantly stratified Mediterranean water column can be established when a slowed down





overturning is assumed. The results also indicate that deposition of halite would take longer than the time span
assumed in the three-stage model. In contrast to the study of (Simon & Meijer, 2017), we use a density driven
dynamic overturning and investigate a broader range of configurations and scenarios. We make our results
transferrable to other semi enclosed basins with an anti-estuarine circulation, e.g. Red Sea (Sofianos & Johns,
2015).

The thermo-haline overturning circulation of this semi enclosed sea is closely linked to its two-way
exchange with the Atlantic Ocean through the Strait of Gibraltar with a dense outflow into the Atlantic, whose
imprints can be traced back to the Tortonian (de Weger et al., 2020). This, in combination with the negative
freshwater budget of the Mediterranean (Simon et al., 2017), strongly implies a simultaneous inflow from the
Atlantic and thus a two-way exchange at the beginning of the MSC. The conversion from Atlantic water to more
saline Mediterranean overflow water happens via an overturning cell in the Mediterranean Sea. This
thermohaline circulation is driven by a combination of convection and sinking events that transport the newly
formed dense water into the deeper basin (Waldman et al., 2018). This process, as well as the interplay between
strait exchange and dense-water formation are still very much the subject of studies, e.g. (Pinardi et al., 2019).

For this type of overturning circulation there are, roughly speaking, two ways to reach a situation where
halite and gypsum are precipitated at the same time. Either the bulk of the Mediterranean Sea has only reached
gypsum saturation while there is a part of the basin that is concentrated in a way that it reaches halite saturation
(henceforth referred to as scenario A), or the basin has reached halite concentration, while some parts stay below
that threshold due to dilution (scenario B).

Scenario A would require an area of deepwater formation where water becomes denser than the water in
the deep basin. There are two different configurations in which this can be achieved. The first one is a basin that
is driven by convection in a distinct part of the basin itself (A1). Alternatively, the overturing cell is driven by
restricted marginal basins from where a density driven downwards flux transports the ions into the deep part of
the basin (A2). In possibility (B) a diluted area is required. It is safe to assume that this would be a marginal basin
with a positive freshwater budget. In that case, the overturning happens via convection between the open and the
deep box. These three configurations as well as their translation to a model set-up are illustrated in Fig. 1.





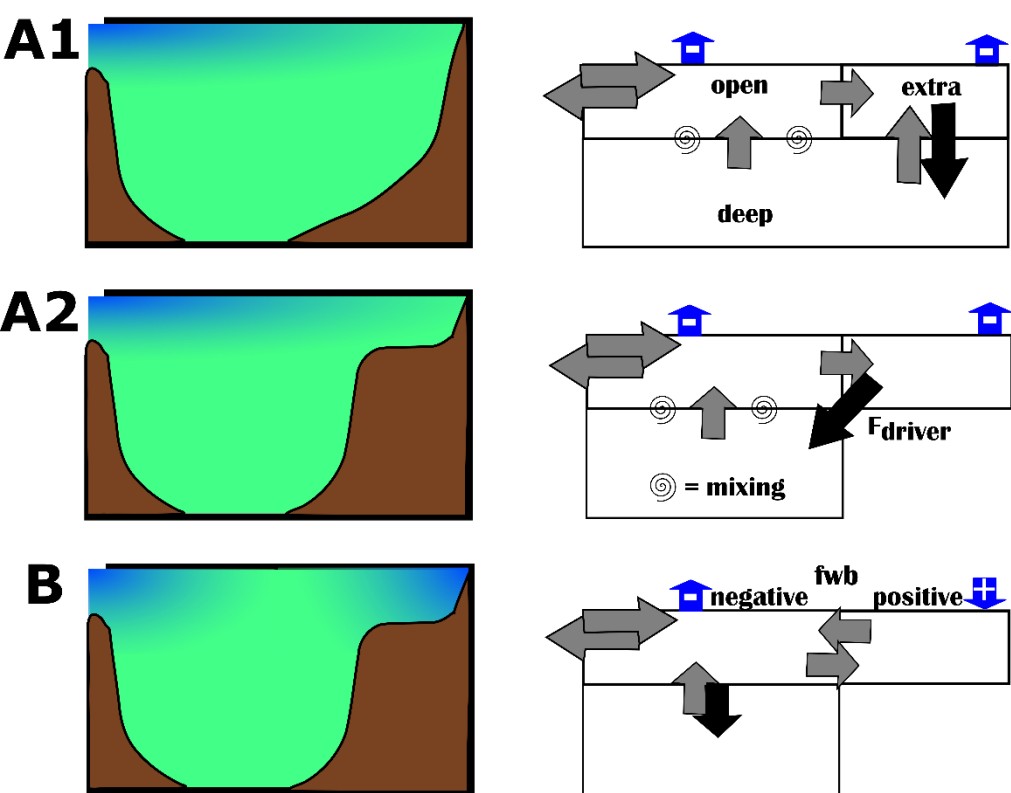

**Figure 1 Visualisation of the three configurations discussed in this paper. Left: The brown outline represents the Mediterranean Sea with the Strait of Gibraltar on the left. Shading symbolises salinity (blue=low, green = high). Right: The corresponding model set-up. The grey arrows indicate the fluxes within the Mediterranean Sea, with the flux that drives the circulation highlighted in black. The vertical blue arrows represent the freshwater budget ($fwb$) of the boxes. The only positive $fwb$ occurs in configuration B in the extra box.**





## 2 Methods

To test whether one of the aforementioned configurations (A1, A2, or B) could lead to coeval precipitation of
halite and gypsum, they were translated to numerical models. From previous studies we know that seemingly
simple conceptual models can help to understand even complex systems, by reducing them to the main processes
and their interaction.

### 2.1 Model

| Symbol | Unit | Value | Explanation |
|---|---|---|---|
| F | $m^3/s$ | | Volume fluxes between boxes |
| $S_{atlantic}$ | $kg/m^3$ | 36 | Atlantic salinity (= initial salinity) |
| $S_{extra}, S_{open}, S_{deep}$ | $kg/m^3$ | | Modelled salinity of the boxes |
| f | - | $0.01 < f < 0.5$ | Relative size of extra box |
| $A_{tot}$ | $m^2$ | $2.5 \cdot 10^{12}$, a | Total surface area |
| $A_{open}$ | $m^2$ | $f \cdot A_{tot}$ | Surface area open box |
| $A_{extra}$ | $m^2$ | $(1 - f) \cdot A_{tot}$ | Surface area extra box |
| $A_{deep}$ | $m^2$ | $A_{tot}$ | Interface area (deep & upper boxes) |
| $V_{extra}$ | $m^3$ | $A_{extra} \cdot 500$ m | Volume extra box |
| $V_{open}$ | $m^3$ | $A_{open} \cdot 500$ m | Volume open box |



| | | | |
|---|---|---|---|
| $V_{deep}$ | $m^3$ | $A_{tot} \cdot 1000\ m$ | Volume deep box |
| $q$ | $m^3/s/\sqrt{kg\ m^{-3}}$ | $10^3 < q < 10^7$ | Restriction parameter |
| $e$ | m/yr | $0.25 < e < 1$ | Net evaporation rate |
| $dt$ | yr | 0.5 | Timestep of the model |
| $c_{A1}$ | - | $0 < c_{A1} < 1$ | Part of $F_{driver}$ not kept in convection |
| $c_{A2}$ | $m^3/s/(kg\ m^{-3})$ | $10^2, 10^4, 10^6$ | Restriction of the margin |
| $c_B$ | $m^3/s/(kg\ m^{-3})$ | $10^2, 10^4$ | Restriction of the margin |
| $e_B$ | m/yr | -0.1 | Net evaporation in margin |
| $\kappa_{conv}$ | m/s | $10^{-1}$ | Scaling parameter for convection |
| $\kappa_{mix}$ | $m^2/s$ | $10^{-4}$ | Mixing parameter |
| $d_{mix}$ | m | 750 | Mixing length |
| $r$ | $-$ | >1 | river water over evaporation |
| $R_q$ | $-$ | | Restriction of basin, determined by outflow and freshwater budget |

**Table 1 Parameters and how they are used in the model. Key to references: a, Meijer (2021)**

For the configurations described in the introduction, the Mediterranean Sea is represented by three boxes. The main part of the basin is divided into two of them, the open and the deep box. The open box represents the surface and intermediate layer up to a certain depth and is thus in interaction with the Atlantic and influenced by the atmosphere through evaporation and precipitation. The deep box represents the deep water and is not directly

influenced by the atmosphere. Those two boxes exchange properties through mixing and they exchange water through density driven fluxes, convection or compensating fluxes, depending on the configuration (see Fig. 1). The so-called extra box describes a smaller volume on the surface that is either a marginal basin (A1 and B) or an



area where convection occurs due to a naturally occurring horizontal salinity gradient (A2). The volumes and area, as well as the other parameters used in this model are listed in Table 1.

All three boxes have a constant volume that is described by their surface or interface area and depth, while their salinities ($S_{open}, S_{deep}, S_{extra}$) are variable. Each flux, $F$, that is triggered by a salinity difference (exchange with Atlantic, $Q$; sinking flux of convection , $F_{open \rightarrow deep}$ or $F_{extra \rightarrow deep}$ ; or dense water sinking) thus triggers fluxes between the other boxes to maintain the volume of each box. All configurations are subjected to a constant net evaporation rate $e$ that acts uniformly across the surface. The only exception is implemented in

B, where the extra box has a positive freshwater budget, which results in a negative net-evaporation rate.

In configuration A1 the driving Flux $F_{driver}$ is triggered when the salinity of the extra box surpasses that of the deep box. A water flux, scaled by a mixing parameter $k_{conv}$, the surface area $A_{extra}$ of the extra box, as well as the salinity difference ($S_{extra} - S_{lower}$), sinks into the deep box. (It is important to note that this work uses the descriptor $\kappa$ not in the traditional sense.)

$$F_{driver} = F_{extra \rightarrow deep} = k_{conv} \cdot A_{extra} \cdot \frac{S_{extra} - S_{deep}}{S_{deep}} \tag{1}$$


and is partially compensated by an upwards flux that is scaled with the factor $c_{A1}$:

$$F_{deep \rightarrow extra} = (1 - c_{A1}) \cdot F_{driver} \tag{2}$$

This creates a convection cell with a net-downwards flux $c_{A1} \cdot F_{driver}$ as it has been described for the Mediterranean Sea (Waldman et al., 2018). The volume of the deep box is then kept constant by an upwards flux

into the open box, and the volume of the extra box is preserved by a compensating flux from the open box that replaces both the freshwater budget ($fwb$) and the net downwards flux. The downwards flux is thus the driver of the circulation. Only the exchange with the Atlantic is not directly dependent on the driver flux. The outflow of this exchange is direct proportional to $\sqrt{S_{open} - S_{Atl}}$ as well as a factor, $q$ ,describing the strait efficiency.

$$Q = \sqrt{S_{open} - S_{Atlantic}} \cdot q \tag{3}$$

The inflow from the Atlantic not only compensates for the outflow, but also for the water volume lost due to evaporation. This creates a stable stratification between the open and the deep box. The resulting salinity gradient leads to mixing at their shared interface. This salt flux $j_{mix}$ , as also used in (Dirksen & Meijer, 2020; Matthiesen & Haines, 2003; Tziperman & Speer, 1994), depends on the salinity difference, the interface area $A_{open}$, a mixing depth $d_{mix}$ and the mixing coefficient $\kappa_{mix}$

$$j_{mix} = \kappa_{mix} \cdot \frac{A_{open}}{d_{mix}} \cdot (S_{open} - S_{deep}) \tag{4}$$






The salinities **S** of the three boxes can hence be described by a set of differential equations using the fluxes **F** as well as mixing the open and the deep box.

$$V_{open}\frac{dS_{open}}{dt} = (Q + eA_{tot})S_{Atlantic} + c_{A1}F_{driver}S_{deep}$$
$$-(Q + c_{A1}F_{driver} + eA_{extra})S_{open} - j_{mix} \tag{5a}$$

$$V_{extra}\frac{dS_{extra}}{dt} = \left((c_{A1})F_{driver} + eA_{open}\right)S_{open} - F_{driver}S_{extra} + (1 - c_{A1}) \cdot F_{driver}S_{deep} \tag{5b}$$

$$V_{deep}\frac{dS_{deep}}{dt} = \underbrace{F_{driver}S_{extra}}_{from\ extra\ box} - \underbrace{(1 - c_{A1})F_{driver}S_{deep}}_{to\ extra\ box} - \underbrace{c_{A1}F_{driver}S_{deep}}_{to\ open\ box} + j_{mix} \tag{5c}$$

In configuration A2 the driving flux also originates from the extra box which here resembles a marginal basin that has restricted exchange with the rest of the Mediterranean. The exchange is again dependent on the salinity difference between the extra and the deep box and is scaled by a parameter for the restriction $c_{A2}$, which changes the flux $F_{driver,A2}$ that is driving the circulation in configuration A2 to

$$F_{driver,A2} = c_{A2} \cdot \left(S_{extra} - S_{open}\right) \tag{6}$$

In configuration B the restricted margin has a positive freshwater budget with a negative net-evaporation rate $e_B$, which makes it fresher than the open and the deep box. Since the extra box here is not producing dense water, its outflow is not the driver of the circulation. Instead, the transport of dense water into the deep here happens via mixing and convection between the open and the deep box, which are thus close to each other in salinity.

## 2.2 Freshened Margin

For a more detailed look at configuration B and to answer the question whether the precipitation of gypsum from such a diluted mixture might be possible, salinity will be viewed as a sum of concentrations.

$$S = \frac{\sum m_{Salts}}{V} = \sum[salts] = [CaSO_4] + [NaCl] + [other\ salts] \tag{7}$$

Where each ion group (NaCl, CaSO₄) has its own saturation concentration which, when exceeded, triggers precipitation (Leeder, 2009; Raad et al., 2023; Topper & Meijer, 2013, 2015). This allows us to take the riverine ion input into the extra box into account and to identify the freshwater budget that would prevent first halite and then gypsum from precipitating in dependency of the evaporation $EP = e \cdot A \ [m^2/s]$ and river input $R = r \cdot$





$EP, r \geq 1$. Expressing the river inflow $R$ in terms of the evaporitic flux $EP$ allows us to formulate the following
expressions in a way that is not dependent on the surface of the basin.

When describing such a state, one can assume that the concentrations of the dissolved salts in
question $[NaCl]_{extra}$ and $[CaSO_4]_{extra}$ as well as the volume do not change over time. Looking at the halite
concentration first, said concentration depends on a balance of sinks (precipitation $\Gamma$ in [kg/s], and ion transport
out of the basin, e.g. $[NaCl]_{extra}F_{out}$) and sources (saline inflow from the Mediterranean $[NaCl]_{open}F_{in}$, low-
saline river inflow $[NaCl]_{river}rEP$. Same can be said for $[CaSO_4]_{extra}$, but for reason of simplicity the
derivation of the final expression will be conducted on the example of halite, which can then be readily translated
to its gypsum counterpart.

$$[NaCl]_{open}F_{in} + [NaCl]_{river}r \cdot EP = \Gamma_{halite} + [NaCl]_{extra}F_{out} \tag{8}$$

With the water volume in the basin being conserved we can express $F_{out}$ as

$$F_{out} = F_{in} + (R - EP) = F_{in} + (r \cdot EP - EP) = F_{in} + (r-1) \cdot EP \tag{9}$$

Using this in addition to the condition that both inflow and outflow are saturated in the salt we are looking at

$$[NaCl]_{open} = [NaCl]_{sat} \quad and \quad [NaCl]_{extra} = [NaCl]_{sat} \tag{10}$$

Eq. 8 can be simplified to

$$[NaCl]_{river} \cdot r \cdot EP = \Gamma_{halite} + [NaCl]_{sat} \cdot (r-1) \cdot EP \tag{11}$$

With $\Gamma_{halite} = 0$ it is possible to identify the point at which precipitation has just not yet started

$$\frac{[NaCl]_{river}}{[NaCl]_{sat}} = 1 - \frac{1}{r} \tag{12}$$

From this we can formulate a condition that applies to a basin that is not yet precipitating halite

$$\frac{[NaCl]_{river}}{[NaCl]_{sat}} < 1 - \frac{1}{r} \tag{13}$$

And, analogously, the upper limit when the basin becomes too diluted for gypsum to precipitate

$$\frac{[CaSO_4]_{river}}{[CaSO_4]_{sat}} > 1 - \frac{1}{r} \tag{14}$$



Which defines a range of values for **r** in which the basin is concentrated enough to precipitate gypsum but diluted enough to not precipitate halite in dependence of the ratio r between river inflow R and net loss to the atmosphere EP.

$$\frac{[NaCl]_{river}}{[NaCl]_{sat}} < 1 - \frac{1}{r} < \frac{[CaSO_4]_{river}}{[CaSO_4]_{sat}} \tag{15}$$

## 2.2 Dimensionless descriptor for restriction

Just like $r$ can be used to compare marginal basins regardless of their size, we can define another dimensionless metric $R_q$ to describe the degree of restriction of a basin with anti-estuarine circulation.

$$R_q = \frac{-fwb}{Q_{out}} \tag{16}$$

This approach is similar but not identical to metrics that have been used by previous studies (Ebner et al., 2024; Flecker et al., 2002; Simon & Meijer, 2015). If $R_q$ is one, then the influx is twice the size of the outflux to the Atlantic, as it has to compensate for an outflow and $fwb$ that are the same size. For a less restricted basin, this ratio is smaller as the basin is more influenced by the properties of the Atlantic inflow. When the basin is more restricted, and the influence of the net-evaporation increases the ratio also increases. This unit-less metric can also be used to compare different basins that are connected via two-way exchange to an oceanic reservoir, regardless of their size.

## 3 Results

In this section the behavior of the three configurations will first be compared to each other. Subsequently they will be analyzed separately to identify conditions that would lead to a concurrent precipitation of gypsum in the margin and halite in the deep. The section ends with a more in-depth look at the last configuration by making a distinction between the concentration of ions related to gypsum and halite.

## 3.1 Comparison

To compare the three configurations to each other, two parameters and their influence on the model need to be discussed. Those are the net evaporation $e$ that acts on the surface, which represents the freshwater budget



expressed as a rate $[m/yr]$, and the strait restriction parameter $q$, which has a somewhat bulky unit $[(m^3/s)/$

$(\sqrt{kg/m^3})]$ that will be omitted further on.

The more restricted the Mediterranean becomes, meaning the less exchange between Atlantic and Mediterranean Sea there is, the higher the average salinity of the system gets. This increase is nonlinear for all configurations, parametrizations, and evaporation rates and only halts when halite saturation is reached. This limit (defined as $S_H = 350\ kg/m^3$) marks the threshold for halite saturation after which salinity would increase

much slower than it did before due to the precipitation of halite.

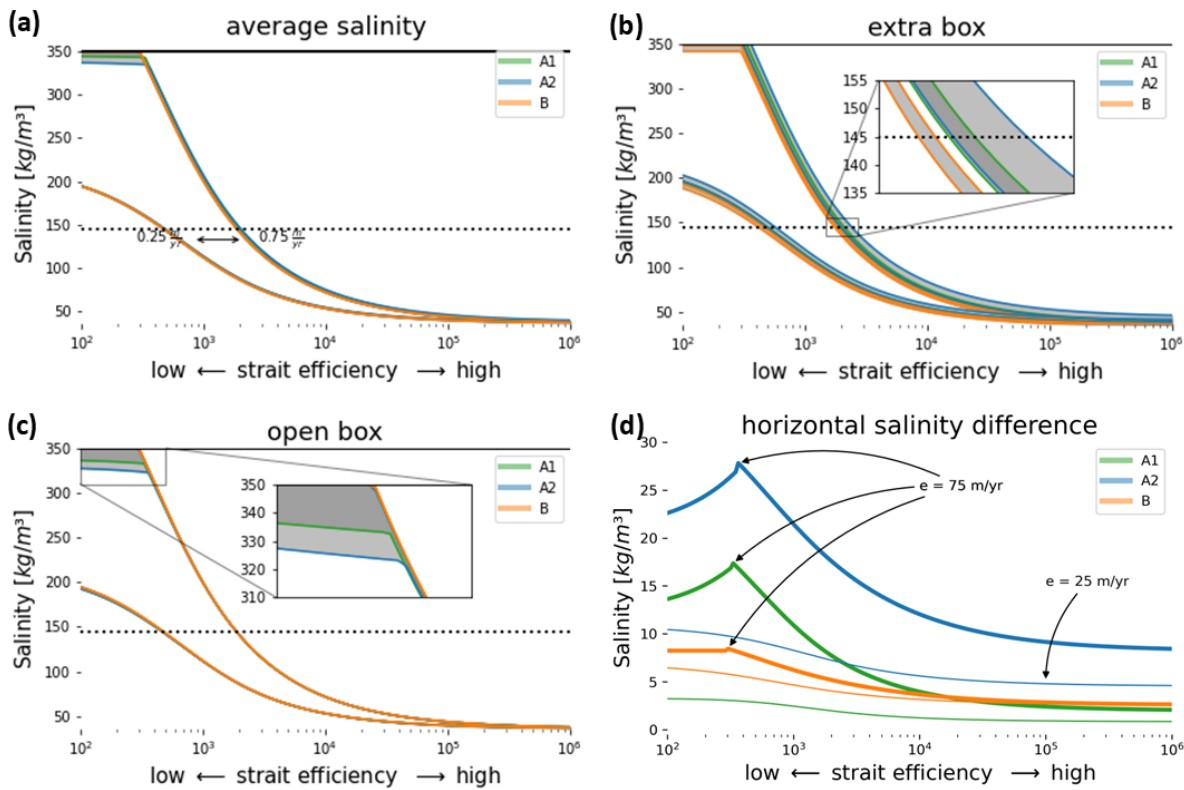

**Figure 2** *Salinity and salinity differences (y-axis) of the 3 different configurations for different degrees of strait restriction (x-axis). Each configuration is run for different sizes of the marginal basin ranging from 1% to 25% f the total surface area. The difference in outcome between the two parametrisation is indicated by the*

*grey area between the two corresponding, coloured lines.*

*Each data point on the lines represents one solution (salinity or salinity difference, y-axis) of the configuration that is defined by a strait efficiency (x-axis). For each configuration there are two sets of lines that only differ in the net-evaporation that was used for the model; a) The average salinity; of the basin b) salinity of the extra box; c) salinity of the upper box; d) salinity differences between the two top boxes, results shown for*





$Area_{extra} = 25\% \ Area_{total}$ ;*Further parameters*: $A1c = 0.1$; $A2c = 10^4 \ (m^3/s)/(kg/m^3)$; $Be_{extra} = -0.1 \ m/yr$, $Bc = 10^4 \ (m^3/s)/(kg/m^3)$

The direction of the horizontal salinity gradient is what differentiates the two A configurations (Fig. 2, blue and green) from the B configuration (Fig. 2, orange). In both A configurations the extra box is the most

saline and thus the first one to reach the halite threshold. In the B case, however, the extra box is diluted compared to the rest of the basin and thus cannot reach halite saturation. The upper and lower box are well mixed in the B configuration. Their salinity develops similarly to the salinity of the extra box in the A configurations.

While the contrast between the average salinity and the constant salinity of the inflow increases, the absolute salinity differences between the boxes also increase, but once halite saturation is reached in the first box,

the salinity differences decrease again. The difference would vanish if the whole system reached halite saturation. For the two A configurations the open box would be the last to reach halite saturation, which only happens when the inflow of the Atlantic gets concentrated to halite saturation by the $fwb$ of the open box. The extra box in the B configuration never reaches halite saturation and thus develops a constant salinity difference to the open box once the open box reaches that threshold. This is a systemic difference between the A configurations and the B

configuration. However, also within one configuration there are differences between runs, depending on the parametrization. Those differences, however, seem small compared to the influence of the forcing (net-evaporation rate $e$).

### 3.2 Analysis

To assess which configurations could lead to coeval precipitation of gypsum in a marginal area and halite

in the deep basin, we will look at each configuration individually and focus on runs where one or two boxes reach halite saturation while at least one stays below that threshold. In contrast to the previous section, the degree of restriction is now expressed as $R_q$. With this metric today's Red Sea would plot at 0.37 ($e_{RedSea} = 2.06 m/yr$, $A_{RedSea} = 4.5 \cdot 10^{11} m^2$, $Q_{RedSea} \approx 0.15 Sv$ (Sofianos et al., 2002)). The Mediterranean Sea is wit $R_Q = 0.11$ slightly less restricted (x in Fig. 3a).





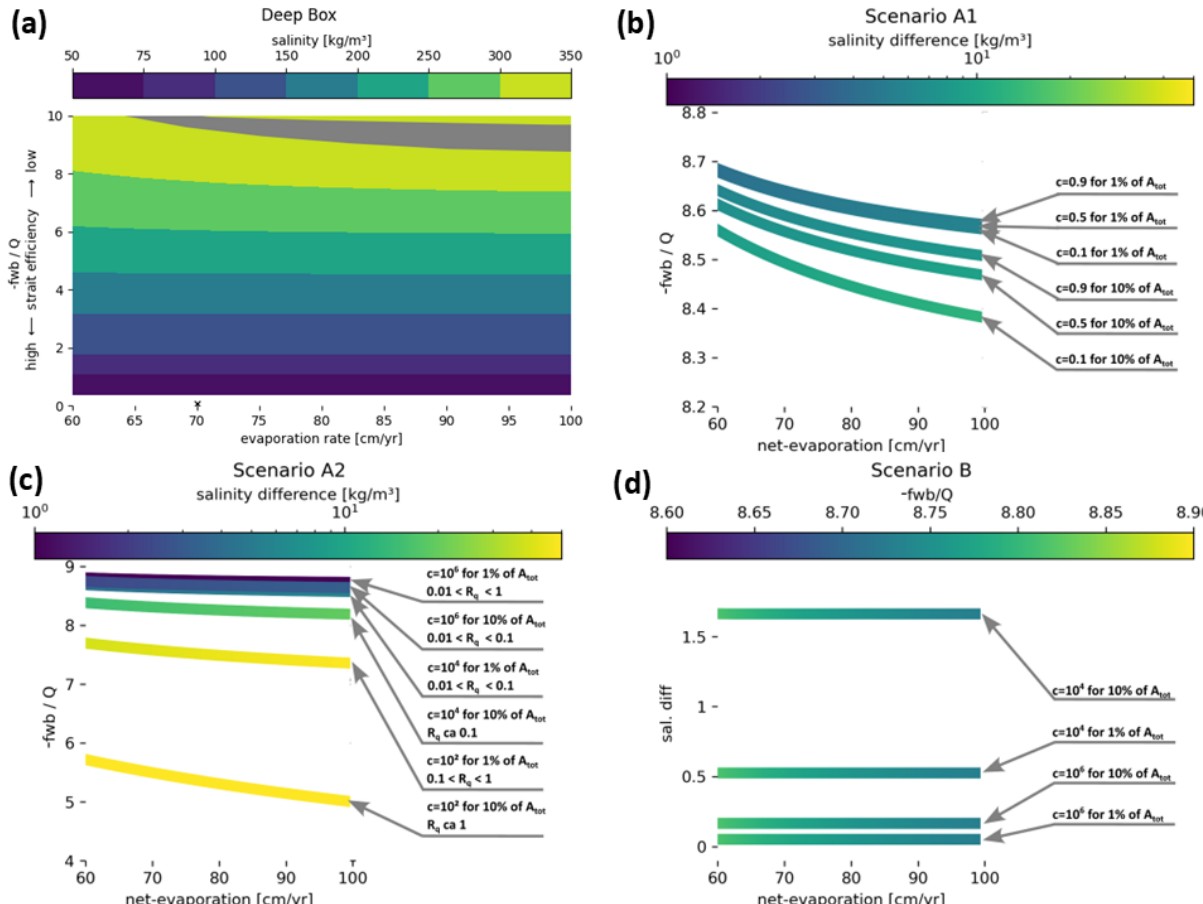


**Figure 3** *Results of the model for a range of net evaporations and degrees of restriction. The x-axis shows the net evaporation rate in $\frac{cm}{yr}$ with increasing values towards the right. The y axis indicates how restricted the system is with respect to the Atlantic (ratio between freshwater budget and outflux). Panels: a) Salinity of the lower box for configuration A1 ($f = 0.9$, $A_{extra} = 0.1 \cdot A_{tot}$). The black x indicates where present day Mediterranean Sea would be located. The grey patch marks where one of the upper boxes has already reached the threshold for halite precipitation, while the other one is not yet saline enough; b) & c) Each line shows the lower limit for patches like the one indicated in a). The colour scaling of the lines shows the salinity difference between the two upper boxes. Notice that the colour scale differs from the one in panel a), and that the y axis is adapted to better show the range of results.; $R_q$ in c) describes the restriction of the marginal basin d) Compared to the previous two graphs, the meaning of the colour scaling and the y axis are switched. Each line now indicates the salinity difference between open and extra box at the lower limit of the patches. The colour scale shows the conditions needed for the system to reach said lower limit*





### 3.2.1 A1: convection

In this convection-driven configuration the halite saturation is reached first in the extra box. Hence, halite
would form there first and then rain into the deep box, while gypsum could simultaneously form in the open box,
which is supplied with "fresh ions" from the Atlantic. This simultaneous precipitation of gypsum and halite in
two different boxes can only occur when the exchange between the Mediterranean Sea and the Atlantic is already
very restricted (grey patch, Fig. 3). For example, for a net-evaporation of $0.6 \, m/yr$ this exchange would need to
be limited to an outflow of less than $5 \cdot 10^{-3} \, Sv$ and for $1m/yr$ ca $8 \cdot 10^{-3} \, Sv$, which is about three orders of
magnitude less than the outflow today (Schroeder & Chiggiato, 2022) for a net-evaporation of $0.7 \, m/yr$. The
lower limit of the grey area indicates the conditions under which the first of the upper boxes reaches halite
saturation, the upper limit indicates the conditions under which both boxes have reached halite saturation. This
situation is reached when the Atlantic inflow is small enough that its less saline water is concentrated to halite
saturation by the loss of fresh water in the open box. The curves in Fig. 3b represent the lower limit of this patch
for different parametrizations of the model, and thus the least extreme conditions with the largest salinity
difference.

When we focus on the range of strait efficiency that would lead to the desired outcome, the influence of
net evaporation stands out. The decline from left to right is caused by the increasing salinity of the basin that
follows stronger evaporation. This means that for lower net evaporation, i.e. going right to left, the basin would
have to be more restricted before the extra box can reach halite saturation. The salinity difference for all tested
parametrizations is much smaller than $50kg/m\text{\textasciicircum}3$, since the extra box is not restricted towards the open box. The
salinity difference increases when more volume is involved in convection (small $c_{A1}$), since that slows down the
circulation between the three boxes. The size of the area in which convection occurs only has a small influence
on the salinity difference. It is thus possible to choose parameter values for configuration A1 in a realistic way
that would lead to coeval precipitation of gypsum and halite, but only when the whole system is already close to
halite saturation due to restricted exchange with the Atlantic.

### 3.2.2 A2: restricted margin

In this configuration halite saturation is also reached first in the extra box which here acts as a restricted
marginal basin. The salinity difference between the two upper boxes mainly depends on the relative strength of
the exchange between the open and the extra box (Fig. 3c), which is comparable to the influence of the restriction





between the Mediterranean Sea and the Atlantic. The extent of this restricted margin also has an influence on this salinity difference, due to the change in surface area that is subjected to evaporation.

Comparing two extra basins with different restrictions but the same surface area (e.g. curves with $A_{extra} = 1\%A_{tot}$ and $c_{A2} = 10^4$ or $c_{A2} = 10^6$, Fig. 3c) shows this more clearly. The loss to the atmosphere is

the same for both for any given net-evaporation, but their outflows differ by two orders of magnitude. This creates a larger horizontal salinity difference for the more restricted basin due to the larger difference between in and outflow. This effect is magnified for a marginal basin with the same restriction but larger area (e.g. curve with $c_{A2} = 10^4$ and $A_{extra} = 10\% A_{tot}$). For such a basin, the salinity difference between the two upper boxes can exceed 50 kg/m³, meaning that the exchange with the Atlantic does not need to be that restricted for the extra

box to reach halite saturation. This, however, is an extreme case and would translate to a marginal basin with the extent of the Aegean Sea (Waldman et al., 2018) and an outflow comparable to the discharge of the Evros [Poulos et al., 2021] that drains into it. Using the same metric for restriction as with the Atlantic-Mediterranean-exchange, this would translate to $(fwb_{marg})/F_{out} \approx 0.02$. To rephrase, it can be said that the less restricted the margin is with regard to its size (low ratio between local $fwb$ and exchange), the closer it is to the overall salinity

of the basin. Which means that for more realistic basins, the Mediterranean Sea would need to be close to halite saturation for the marginal basin to reach that threshold.

### 3.2.3 B: freshened margin

No matter the size or restriction of the extra box (freshened margin), the open box reaches halite saturation always for the same parametrization (Fig. 3d, color scale), while the extra box always stays below that

threshold. A plot like in Fig. 3a for this configuration would thus not show an upper end for the grey patch. This is due to the positive local freshwater budget, $e_B = -0.1 m/yr$, in the extra box that dilutes the influx from the open box. The resulting salinity difference depends on the relative restriction but would also, just like in the previous configuration, all runs.

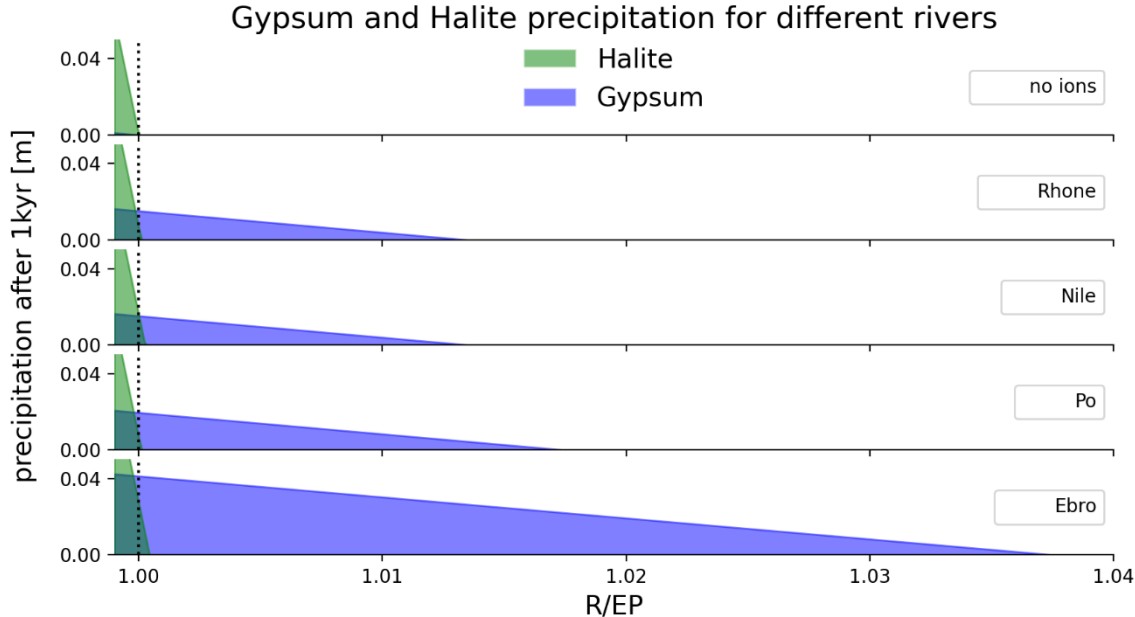

**Figure 4** *Gypsum and halite precipitation in a margin that is freshened by river water and fed by the Mediterranean Sea which is saturated in gypsum and halite. The x axis shows the ratio between the river inflow and evaporation (corrected for precipitation). When this ratio is 1(black dotted line), the fwb is 0. For R/EP>1 the basin experiences less evaporation than river input and has a lower salinity than the inflow from the Mediterranean Sea. The y-axis shows the precipitation rate that would result from those conditions in m/kyr*

In contrast to the previous two configurations, looking at salinities is not enough to determine whether gypsum would precipitate. The salinity in the freshened margin might be above the gypsum threshold, but since it is a diluted brine, it might not be saturated in gypsum anymore. To determine whether that is the case, a closer look at the chemistry of the brine is needed. Since the extra box describes a generic unspecified marginal basin and the chemical composition of the generic river into that basin is not known, the closest approximation is investigating the behavior of the margin if it was diluted by typical Mediterranean rivers (Fig. 4).

If the river would carry no ions, then a positive *fwb* would prevent precipitation of gypsum and halite since the brine would become undersaturated in both minerals. If the river water, however, also brings in ions, one or both concentrations could stay at this threshold and surplus ions would precipitate. This is why for river water compositions closer to the one of Rhone, Nile, Po or Ebro (Gaillardet et al., 1999), gypsum could also precipitate when $R/EP > 0$.



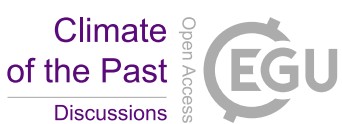

The results in Fig. 4 show two things. Firstly, that it is theoretically possible to precipitate gypsum from such a diluted margin without also precipitating halite, and secondly, that the total inflow from rivers can only exceed the loss of fresh water by 0.01% to 4% depending on the concentration of $Ca^{2+}$ and $SO_4^{2-}$ ions in the river water. The smaller the fwb compared to the surface area is, the higher the precipitation rate. The absolute rate depends on the evaporation rate but is in the order of magnitude of $0.1 m/kyr$.

## 4 Discussion

### 4.1 Limitations of the model

The models presented here not a representation of the complexity of the Mediterranean Sea, but rather a means to understand some aspects of it. As such it is important to be aware that the dynamic between eastern and western Mediterranean is not included in the different configurations and that the way salinity is treated does not capture the full thermodynamic reality of brines. Precipitation of halite in the deep basin, for example, is not included in this work, since the salinity cannot increase without the influence of evaporation, and the threshold for precipitation in this work is not dependent on the chemical composition of the brine, pressure, or temperature.

A lower temperature also decreases the ability of the waterbody to dissolve ions by $0.22 kg/m\verb|^|3$ per$1°C$ (based on water chemistry of the Dead Sea, (Stiller et al., 1997). This is the cause of one of the governing processes of the Dead Sea (Sirota et al., 2016) and could also play a role in the Messinian Salinity Crisis, since the present Mediterranean shows a vertical salinity gradient (e.g. (Fach et al., 2021; Margirier et al., 2020)). In the model this would have the effect that Halite saturation might be reached first in the deep basin and that precipitation could also occur in the deep basin. The ions for this would be provided by the warmer downflow, which would have an excess of ions when the saturation concentration gets lowered due to cooling when the sinking water mass mixes with the saturated colder one. In the extreme case, we can estimate the precipitation rate caused by that process, by assuming a strength for the downwards flux, $F_{down}$, and temperature difference, dT, (see Table 2). With those assumption it is possible to calculate the ion stream that would be in excess by cooling down the downwards flux

$$j_{excess} = F_{down} \cdot \frac{dc_{sat}}{dT} \cdot dT_{down} = 2.2 \cdot 10^5 \frac{kg}{s} \tag{17}$$

Distributing $j_{excess}$ over the total surface area of the Mediterranean would lead to a sedimentation rate of $10 m/kyr$. This is in the same order of magnitude as ions added via an inflow with marine composition





$(c_{NaCl,Atl} = 2.7 kg/m^3$ ,(Leeder, 2009)) to a basin with $e = 1 m/yr$ and a restriction that leads to halite saturation
(Fig. 3)

$$j_{added} = Q_{in} \cdot c_{NaCl,Atl} = \left( \frac{e \cdot A}{\left( \frac{fwb}{Q} \right)} + e \cdot A \right) \cdot c_{NaCl,Atl} = 2{,}14 \cdot 10^5 \frac{kg}{s} \tag{18}$$

| variable | value | | variable | value |
|---|---|---|---|---|
| $F_{down}$ | 1 Sv | | $\rho_{halite}$ | 2300 kg/m³ (a) |
| $dT_{down}$ | 1°C | | $A_{sediment}$ | $2.5 \cdot 10^{12} m^2$ (b) |
| $R$ | 8.2 | | $e$ | 1 m/yr |

**Table 2** *Assumptions for calculation of ions flux from one saturated box to another in dependence of their temperature difference.* **Key to references: a, (Leeder, 2009); b, (Meijer & Krijgsman, 2005)**

It is however questionable if such a thought experiment is not a simplification of a complex process, since the heat of the cooling stream is dissipating into the surrounding waterbody, raising its temperature and saturation concentration just enough to take the excess ions in. On a smaller scale however, the dependency of the saturation concentration on temperature can indeed lead to substantial halite deposits Sea (Sirota et al., 2020). Those double diffusive processes like salt fingering are too complex (Ouillon et al., 2019) be represented reliably in this type of model.

Another simplification is the use of constant evaporation rates, in contrast to a forcing that reflects the changes in the freshwater budget over time. Studies on the formation of sapropels in the Black Sea, which use a comparable approach, show that the transient response of such a model to changes in the forcing can be complex (Dirksen & Meijer, 2022; Dirksen & Meijer, 2020). Another study, using a version of configuration A1, however showed that a sinusoidal freshwater budget influences the amplitude depending on the restriction, but not the average over time of signals like salinity (Ebner et al., 2024).

A1 is also similar to the model used in (Simon & Meijer, 2017). The main difference is in the definition of the driving flux and the exchange with the Atlantic. While their model is defining with set values, the model presented in this paper is scaling them more dynamically with salinity differences. Another major difference is the assumption that halite also precipitates from a waterbody that is not in contact with the atmosphere. While this would be allowed in the model presented here, it is not possible since no flux going into the deep box is above the threshold of saturation.





## 4.2 Implications of the model results

As elaborated in the results section of this paper, all three configurations could lead to a situation in which halite and gypsum would form simultaneously, but spatially separate. While the conditions under which this would happen seem similar (high overall salinity), they do differ from each other. This is best presented by their placement on a theoretical timeline of a hypothetical basin that experiences salinification. If we imagine this basin to be at gypsum saturation, it would develop in three steps as salinity keeps increasing.

**Step 1;** A restricted margin, with a salinity that is higher than the average reaches halite saturation and starts precipitating halite. When the salinity of the main basin increases further, halite also starts forming in other, less restricted marginal basins.

**Step 2;** The salinity in the open basin is now so high, that also unrestricted areas, reach halite saturation. Those are the interplay of horizontal and vertical salinity gradient leads to density instabilities and thus convection. The crystals start forming there and rain into the deep where they might partially dissolve again (Topper & Meijer, 2013)

**Step 3;** The basin has now reached halite saturation and halite is the predominant evaporite that forms and rains into the deep. Only some marginal areas, that are experiencing a local positive freshwater budget are still precipitating gypsum without halite. This gypsum is now mainly influenced by the chemistry of the river water and how the river inflow compares to the evaporation that occurs on the surface.

The situation in step 1 would lead to halite deposits in sub-basins that only have restricted exchange with the rest of the basin. While it might be possible that the dense, saturated water, leaving such a sub-basin, would form halite as it sinks into the deeper part of the basin and cools, this process would be hindered by the mixing with other undersaturated water masses. While this might happen locally it is unlikely to be a mechanism that forms significant amounts of halite in deeper parts of the basin, where most halite deposits are found in the Mediterranean Sea, with only few exceptions in elevated basins in for example the Balearic Promontory (Heida et al., 2022; Raad, 2022; Raad et al., 2023; Raad et al., 2021). Configuration A2 seems to explain the halite deposits in the Balearic promontory which hosts marginal basins. However, a case study on one of these, the Central Mallorca Depression (Raad, 2022), showed that its halite deposit was most likely caused by a draw down, and not by a sill restricting the exchange. In step 3, only miniscule amounts of gypsum would be formed in the marginal areas. The sedimentation rates that would result from such a mechanism are comparable to those that result from the thickness of the lower Tripoli Unit in the Lorca basin. This rate has been explained, however, by a gap in sedimentation that makes it impossible to define the age of the base of the unit (Rouchy et al., 1998). Even under





perfect conditions a sedimentation rate of $< 4 cm/kyr$ would take more than 25 kyr to deposit one meter of gypsum. Other estimates for gypsum deposition rates exceed this value by several orders of magnitude (see Table 3).

| sedimentation rate | location | time |
|---|---|---|
| $100 - 1000\ cm/kyr$ | salinas in Spain | Present (a) |
| $20\ cm/kyr$ | E. Spain, N. Apennines | Late Messinian (b) |
| $8000\ cm/kyr$ | Shallow margins | Late Messinian (c) |

**Table 3 Gypsum precipitation in different studies. Key to references: (a) (Manzi et al., 2012); (b) (de Lange & Krijgsman, 2010) ; (c) (Lugli et al., 2010)**

The most interesting step in terms of likelihood and significance of the synchronicity of halite and gypsum thus seems to be the one that corresponds to configuration A1 and is described in Step 2. It covers the transition between basin wide gypsum and basin wide halite precipitation that does not depend on local factors,
only a horizontal salinity gradient, which unquestionably exists in the Mediterranean Sea (Bonnet et al., 2013). In the context of the Messinian Salinity Crisis this process would add some time to halite formation, as phases 2 and 1 of the consensus model (Roveri et al., 2014) might overlap. How much time depends on the strength of the salinity gradient and the pace of the salinity increase. Since the salinity of the Mediterranean would react sensitively to small changes in restriction once it is restricted enough to be close to halite saturation (Meijer,
2012), this overlap likely was insignificantly short.

**5 Conclusion**

This study allows us to explore the different configurations that could have led to a simultaneous precipitation of gypsum and halite. We find this to potentially occur in all configurations, but only for average salinities close to halite saturation. Based on this we propose a timeline for a salinifying basin with restricted margins, where one of
the configurations describes the transition between predominantly halite and gypsum precipitation and the other two configurations might have been local effects occurring just before and after this transition, but not to a degree that it mayorly influenced the MSC strata. Our results do not exclude the possibility of an earlier onset for halite precipitation in the eastern sub-basin.





## 6 Code and Data availability

The python scripts for the models and the analysis of their results are made available. The repository under

DOI:10.5281/zenodo.12511228 ({OpenAIRE}, 2013) also includes the results of finished runs that are presented in this

work.

## 7 Author contributions

The models and their analysis were written and conducted by Ronja Ebner, the interpretation was developed and discussed

closely with Paul Meijer.

## 8 Acknowledgements

This work was conducted in the context of SALTGIANT, an European project funded by the European Union's Horizon 2020

research and innovation programme under the Marie Skłodowska-Curie grant agreement n° 765256.

## 9 Competing interests

The authors declare that they have no conflict of interest.

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
