# Peer review of "A question of time and space: A model approach to the synchronous precipitation of gypsum and halite during the Messinian Salinity Crisis"

_Climate of the Past, 2024_

## Referee Comment (RC1)

**Reviewer comments on:**

*Ebner and Meijer (under review in Climate of the Past, 2024)*

*A question of time and space: A model approach to the synchronicity of gypsum and halite during the Messinian Salinity Crisis*

**Summary**

Despite more than five decades of research, several mysteries related to the Messinian Salinity Crisis (MSC) remain unresolved. One such controversy is the order of deposition of different units in marginal versus deep basins of the Mediterranean (for example, Primary Lower gypsum/ Upper gypsum units in the marginal basins. Halite-Resedimented lower gypsum and Upper gypsum units in deep basins – Roveri et al., 2014). Ebner and Meijer employ simplified box modeling to test whether gypsum and halite could have been simultaneously precipitated in marginal and deep basin settings during the MSC, rather than during different stages. Their thought experiments are carried out with 3 basin configurations (A1, A2 and B).

Each configuration includes 3 main boxes (open, extra and deep), where the extra box represents a restricted marginal basin in A2 and B configurations. The numerical model design incorporates volume fluxes within the basins, two-way exchange with the Atlantic, a freshwater budget and circulation mechanisms including convection and vertical diffusion. These processes are then employed in calculating salinity evolution of different boxes. The dependency of salinity evolution of the boxes on Gibraltar Strait restriction effect, freshwater budget and volume kept in convection are evaluated subsequently.

I appreciate the effort put into this thought experiment. Such simple models may provide starting points for further development of more complex models in future studies. However, I think there are a few aspects that need to be improved, prior to publication.

Below, I in detail explain my major and minor concerns.

**Specific comments:**

1.  A proper comparison with the real-world scenario
    The aim of the authors is to test if their simplified model predicts whether the salt units (gypsum and halite) in marginal and deep basins could have been precipitated concurrently. Even though they briefly explain the existing hypotheses for the depositional patterns within the MSC, I suggest that proper definitions of marginal *versus* deep basins should be provided prior to elaborating their experiments, with suitable diagrams of existing models for depositional units in each setting (for example, a simplified version of Roveri et al (2014) synthesis). Such insight would make it easier for the reader to follow the author's intentions, in relation to their model experiments. I propose that the same diagram may be used to show the ambiguity in horizontal continuity of the PLG, as the authors indicate in their introduction.

2.  Organization
    Added to my above suggestion, the manuscript may be ordered in the following sequence: A general introduction to the MSC, with explanations on marginal versus deep basins; Existing hypotheses for the different timing of sedimentary unit deposition (including diagrams); Methods; Results; Discussion – here, I suggest including a better explanation of their model results with respect to actual observations they provided in the revised introduction. Under implications, the authors combine their different configurations to develop a timeline of salinification. I suggest adding a new diagram to explain their timeline, as this is one of their final interpretations, and Conclusions.

3. Timescales

A majority of MSC researchers suggest that the PLG unit was developed as an alternating sequence of gypsum-marl couplets, with up to 17 units paced by insolation (Lugli et al., 2010, Manzi et al., 2013). For someone who may try to compare the suggested scenarios with existing timescales of PLG/ Halite unit deposition (eg: PLG stage during 5.97-5.60 Myr, Halite deposition during 5.60-5.55 Myr), no information has been provided regarding the timescales considered for model experiments.

For instance, it has not been shown how long it will take to reach halite saturation in the extra box in A1 scenario. Provided that, the extra box should precipitate gypsum before reaching halite saturation. What are the periods required to reach gypsum and halite saturation points? How do they compare to the suggested insolation-paced alternations for marginal basins? How would the strait efficiency parameter impact these timescales?

Unless I'm mistaken, such information does not exist in the present manuscript. Because the authors are aiming to relate their experiments with existing hypotheses, I suggest that these comparisons would be important. Is it possible to add a brief explanation of these to the manuscript?

**Technical corrections and minor comments:**

Title- *A question of time and space: A model approach to the synchronicity of gypsum and halite deposition during the Messinian Salinity Crisis.* Suggestion to add 'deposition' to the title.

Line 4 – Earth Sciences – 2 words

Line 6 – well studied….

Line 7 – define Ma

Line 11 – …for different configurations… of what?

Line 17 - …a sufficiently restricted marginal basin….

Line 18 – remove 'the one of' and 'areas of the' → gives- …once the average salinity approaches halite saturation it can also form in the open basin…

Line 19 – same as for Line 17

Line 20 – define kyr

Line 21 - …within a one basin… rephrase

Line 27 – change to …youngest salt giant formation…

Lines 28-29 – suggestion to reorganize for clarity

Lines 47-48 – vague statement …a more recent study, however, reopened this question again…

Figure 1 – Add labels of different parameters (eg: evaporation, convection, diffusion) to one of the configurations. Strait of Gibraltar is not properly visible as you have not shown the Atlantic side.

Line 95 – add citations: From previous studies, we know…

Model symbols and parameters – If the relative size of the extra box is *f*, shouldn't $A_{open}$ be *(1-f)A_{tot}* and $A_{extra}$ be *(f)A_{tot}*?

For $V_{extra}$, you have not prescribed what 500 m is (which, I assume is the depth/ thickness of the water column)

Line 135 – To arrange the diffusivity term in order → suggestion to rearrange the equation to

$$j_{mix} = K_{mix} \cdot (S_{open} - S_{deep}) \cdot \frac{A_{open}}{d_{mix}} \cdot$$

Line 140 – Shouldn't the salt flux be upward, therefore for equations 5a and 5c the $j_{mix}$ terms become opposite in sign (positive for 5a and negative for 5c)?

Lines 156 – should you mention the Black Sea as well?

Figure 3c – Suggestion to update the label ca to $ca_2$ (correct?)

Line 271 – can you state the salinity difference?

Line 274 – You have not stated to which figure you are referring to.

Lines 283 – perhaps move 'also' in front of 'halite'?

Line 300 – Reads disorganized when you start the sentence with 'which'.

Line 362 – Should 'Sea' be removed?

Line 362 – Reads disorganized when you start the sentence with 'those'.

Lines 363 - …to be represented?

Line 383 – suggestion to add a figure explaining you time series of salinification.

Lines 388 and 391 – into the deep basin?

Line 427 – majorly?

Line 428 – Explain why your results do not exclude this (vague statement).

---

## Referee Comment (RC2)

Review of the Manuscript

A question of time and space: A model approach to the synchronicity of gypsum and halite during the Messinian Salinity Crisis

Ronja M. Ebner, Paul Th. Meijer

The formation of salt giants, such as the Messinian Salt Giant during the Messinian Salinity Crisis (MSC, 5.97–5.33 Ma) in the Mediterranean Sea, remains a topic of ongoing scientific debate. Traditionally, it has been assumed that gypsum precipitated in the marginal areas before halite formed in the deeper basin, although this has not been definitively confirmed.

This study uses box modeling to explore different halite and gypsum distribution scenarios. Results show that, under certain conditions, both minerals could precipitate simultaneously but in different areas. The authors propose that halite may form in restricted margins as salinity increases before spreading to open areas when salinity reaches halite saturation. This modeling approach offers valuable insights into the dynamics of evaporite formation in semi-enclosed basins like the Mediterranean Sea during the MSC.

The manuscript presents a structured and well-thought-out investigation of the factors influencing gypsum and halite precipitation in the Mediterranean Sea. It adds to our understanding of evaporite formation and, with further refinement, could contribute significantly to the field. However, I believe this work might be more suited to a journal like *Geoscientific Model Development (GMD)*, focusing on model description and validation.

**A few questions:**

While the authors suggest that the model could be applied to other basins (e.g., the Red Sea), it is not clear how the specific model configurations (A1, A2, B) would translate to different geochemical settings. Could the model be adapted to explore other evaporite-forming basins more explicitly?

Regarding the title, you should mention the "Mediterranean Sea" because the MSC occurred in the Med Sea, and your study focused on the Med Sea.

The paper mentions that constant evaporation rates were used. How might a variable evaporation rate, could impact the model results? Could this change the timing or locations of gypsum and halite precipitation? Were there any sensitivity tests performed to explore this?

The manuscript does not provide sufficient discussion on the role of the Strait of Gibraltar in influencing Mediterranean circulation and salinity. A more detailed analysis of how restricted or variable water exchange through the Strait affects gypsum and halite precipitation patterns would add depth to the study.

Have you conducted sensitivity tests on key parameters such as evaporation rates, river water composition, or Strait of Gibraltar exchange? If not, how might these factors impact your results?

Missing punctuation occurs in multiple sentences where commas or periods could help separate clauses or clarify meaning (references style, the caption of the figure in bold, the table legends …).

**Abstract**

The abstract could benefit from a clearer articulation of the novelty of the study. It touches on known issues but doesn't strongly emphasize how the modeling results diverge from or contribute to existing theories.

**Introduction**

The introduction references key studies and models that have addressed the MSC. The mention of studies that confirm or question these models (e.g., Meilijson et al., 2019; Manzi et al., 2018) highlights the ongoing scientific debate and the gaps in current understanding. The comparison with Simon & Meijer (2017) is helpful, but the contributions of the present study (e.g., density-driven dynamic overturning) could be more explicitly emphasized early on. For instance, the detailed breakdown of different studies (e.g., Meilijson et al., 2019 vs. Manzi et al., 2018) could be summarized more concisely to avoid overloading the reader with too many specific comparisons at the outset.

Citations are included in parentheses, but in some cases, they interrupt the flow of the text. For better readability, consider rephrasing sentences to integrate citations more naturally. Example: Instead of "5.97 to 5.33 Ma, (Roveri et al., 2008)," you could say "According to Roveri et al. (2008), the event occurred between 5.97 and 5.33 Ma." This would make the text smoother.

Consistency in citation formatting is needed. For example, in some instances, authors' names are written in all caps, which should be corrected., e.g. (Decima & WEZEL, 1971; Decima & Wezel, 1973)

The flow between ideas could be improved with clearer transitions between sections. For example, when moving from the discussion of modeling to the thermo-haline circulation section, adding transitional sentences can help guide the reader more smoothly from the background after the modeling approach.

The conversion from Atlantic water to more saline Mediterranean overflow water happens via an overturning cell in the Mediterranean Sea." Not clear, this sentence could be rephrased.

The abbreviation "MSC" for Messinian Salinity Crisis is introduced but not consistently used throughout the text. It would help to use the abbreviation after it's introduced to avoid repeating the full term, e.g. line 342.

**Method section**

The overall structure and technical content are strong, but enhancing transitions will improve readability.

While you define many variables, key terms could be better explained to ensure the reader fully understands. For example, explaining "net evaporation rate" in more detail would help if a reader is not familiar with the exact context. Similarly, more context around κ and why it's used differently from its traditional sense could be provided upfront to avoid confusion.

Some terms such as "anti-estuarine circulation," "driver flux," and "marginal basin" are used without sufficient context for non-expert readers.

After describing each configuration (A1, A2, and B), it might be helpful to summarize their key differences in a table. This would help the reader quickly differentiate between them.

What is the temporal resolution of your model, and how does influence the results, particularly regarding the timing of halite and gypsum precipitation?

**Results section**

This section is a well-structured and detailed examination of the different box model configurations, demonstrating the complexities of salinity dynamics in semi-enclosed basins like the Mediterranean. The authors have done a commendable job breaking down the influences of key parameters, such as net evaporation ($e$) and strait restriction ($q$), and their effects on the system's salinity gradients and mineral precipitation.

The use of the strait restriction parameter ($q$) and its bulky unit [$(m^3/s)/(\sqrt{kg/m^3})$] is well justified, but simplifying its interpretation would help make the section more accessible.

The model uses generic assumptions about river water composition to assess gypsum precipitation in the extra box. How significant are variations in river chemistry (e.g., calcium and sulfate concentrations) for altering the results, and were sensitivity tests performed with different river compositions?

The section compares the model results with the Mediterranean and Red Seas, I think that the appearance of the part about the Black Sea is very abrupt, and there is very little information about the Black Sea in the paper.

**Discussion section**

The discussion is rich in technical detail but sometimes lacks a clear "so what?" moment that emphasizes why these results are significant in the context of the Messinian Salinity Crisis or other studies on evaporite formation.

While the model's limitations are well discussed, it would be helpful to suggest what future studies could address based on these results. How could the model be improved? What future work is needed to fill the gaps identified in your study?

**Conclusion section**

The conclusion, while summarizing the key findings, could be strengthened by tying the results more explicitly to potential future research directions or practical implications. It currently ends somewhat abruptly and could benefit from a more definitive closing statement on the significance of the study.

For example, what does this timeline and model tell us about the general understanding of evaporite formation in restricted basins? How might these findings inform future models or field studies in similar settings?

**A few grammatical errors and missing punctuation marks**

There are a few grammatical errors and missing punctuation.

Legend of figures in bold?

Ensure that table legends appear at the top of the tables. This would align the manuscript with common publication standards.

A few suggestions:

Line 6 "Saltgiants" => "Salt giants" ?

Line 9: "could be not yet been confirmed" => "could not yet be confirmed".

Line 32: "reaches up to three km" => "reaches up to three kilometers".

Line 40: "unambiguous, since we, for example, cannot follow" should be "unambiguous since, for example, we cannot follow".

Line 48: "re-opened" => "reopened".

Line 80: "overturing" => "overturning".

Line 334: "The models presented here not a representation of the complexity..." => "The models presented here are not a representation..."

Line 340:  a space between "per" and "1°C" (per 1°C)

Line 427: mayorly?

---

## Referee Comment (RC3)

A question of time and space: A model approach to the sychronicity of gypsum and halite during the Messinian Salinity Crisis

Ronja Ebner and Paul Meijer

This manuscript uses a box model to explore the circumstances under which halite and gypsum could form at the same time in a marginal marine setting. In doing so, the paper addresses a question that has remained unanswered over the last 50 years, since late Miocene evaporites were first identified below the Mediterranean's sea floor with some recent papers on the Messinian Salinity Crisis advocating the synchronous precipitation of halite in the deep basin and gypsum on the margins and other advocating asynchronous precipitation. The three scenarios presented are clearly presented and the modelling, though simple, provides a powerful evaluative tool allowing the authors to conclude that while synchronous precipitation is possible, none of the three scenarios can realistically account for the volumes of evaporites preserved today.

I find the story that has been presented and the model results compelling however there are a couple of issues that I would like the authors to address both to clarify the current focus and to push their model-based thinking a little further than the paper does at present:

1. The box model construction illustrated in Figure 1 shows two-way Mediterranean-Atlantic exchange. It is widely accepted that this configuration probably only applies to Stage 1 of the MSC, when gypsum was precipitated in the marginal basins of the Mediterranean requiring a high sea-level. Stage 2 and 3 are more likely to have occurred with Atlantic inflow but negligible outflow from the Med, consistent with a base level that was below the level of the gateway. Consequently, the main application of this model configuration is Stage 1. This is mentioned in the abstract but is not made clear in the introduction where a description of all three phases of the MSC (L31-39) is followed by a statement about the challenges of shallow-deep water correlation as a justification for looking at synchronous gypsum-halite precipitation (L50-50).

2. The paper concludes that synchronous precipitation of gypsum and halite can only happen in Scenario A when the system as a whole is close to halite saturation. While I accept the statement in the first paragraph of the discussion (L334-339) that the model is not meant to represent "the complexity of the Mediterranean Sea", none the less, it is possible at least to point out the episodes within the MSC that are closet to the model configuration used and consider the implications. For example, some discussion about when within Stage 1 reaching near halite saturation is most likely would enhance the applicability of the results. The strait efficiency required to generate synchronous gypsum-halite precipitation could be evaluated against the Sr isotope ratio data for Stage 1 which progressively diverges from the ocean water curve. This might then enable them to evaluate the duration of the potential overlap between Stage 1 and 2 mentioned in L417-20.

3. Section 3.2.3 (Scenario B) – this section needs a little more explanation of the chemistry and particularly some more information about the chemistry of the rivers that are modelled in Fig. 4 so that the reader can see how their different compositions result in different consequences.

In terms of technical issues, the manuscript could be improved in the following ways:

- Title – suggested tweak "A model approach to the synchronous precipitation of gypsum…."
- Abstract L9 – "different configurations". A little more clarity about what those configurations are would help here
- Abstract L16 – "salinifying". Suggested alternative "a timeline for an increasingly saline basin."
- Figure 2d – for clarity, label the y-axis "salinity difference"
- Figure 3 – both the caption and the text in line 244 state that there should be an "x" showing the present-day Mediterranean on Fig 3a, but I can't see it.
- Discussion L362 – the word "Dead" is missing before "Sea"
- Section 4.2 L405 – the thickness of the lower Tripoli Unit in the Lorca basin is used to illustrate the likely sedimentation rates resulting from step 3, but that isn't helpful if you don't know how thick that unit is…. and I don't!

---

## Author Comment (AC1)

**Dear reviewer 1,**

**we would like to thank you for your helpful and thoughtful comments. To address them properly we chose to answer them one by one.**

*1. A proper comparison with the real-world scenario*
*The aim of the authors is to test if their simplified model predicts whether the salt units (gypsum and halite) in marginal and deep basins could have been precipitated concurrently. Even though they briefly explain the existing hypotheses for the depositional patterns within the MSC, I suggest that proper definitions of marginal versus deep basins should be provided prior to elaborating their experiments, with suitable diagrams of existing models for depositional units in each setting (for example, a simplified version of Roveri et al (2014) synthesis). Such insight would make it easier for the reader to follow the author's intentions, in relation to their model experiments. I propose that the same diagram may be used to show the ambiguity in horizontal continuity of the PLG, as the authors indicate in their introduction.*

**We agree that adding a figure to visualize the two different options we are exploring would be beneficial for the reader. We chose to do this in a way that relates directly to the structure of our model (first draft see below).**

[Figure]

| |
|---|
| **Alternative scenario of synchronous precipitation of gypsum and halite** / **Consensus model** |
| **In the consensus model the end of gypsum precipitation in the margins coincides with the begin of halite precipitation in the deep basin. We are exploring an alternative scenario with those two processes overlapping in time.** |

*2. Organization*
*Added to my above suggestion, the manuscript may be ordered in the following sequence: A general introduction to the MSC, with explanations on marginal versus deep basins; Existing hypotheses for the different timing of sedimentary unit deposition (including diagrams); Methods; Results; Discussion – here, I suggest including a better explanation of their model results with respect to actual observations they provided in the revised introduction. Under implications, the authors combine their different configurations to develop a timeline of salinification. I suggest adding a new diagram to explain their timeline, as this is one of their final interpretations, and Conclusions.*

**We also prepared a figure to visualize the timeline (see below) as described in line 383 - 393. This figure in combination with the additional discussion of the time component (see question 3) will add depth to the discussion.**

[Figure]

**Proposed timeline of a basin with increasing salinity (left to right). The yellow patches describe the location of the halite deposits formed at each step.**

*3. Timescales*

*A majority of MSC researchers suggest that the PLG unit was developed as an alternating sequence of gypsum-marl couplets, with up to 17 units paced by insolation (Lugli et al., 2010, Manzi et al., 2013). For someone who may try to compare the suggested scenarios with existing timescales of PLG/ Halite unit deposition (eg: PLG stage during 5.97-5.60 Myr, Halite deposition during 5.60 5.55 Myr), no information has been provided regarding the timescales considered for model experiments. For instance, it has not been shown how long it will take to reach halite saturation in the extra box in A1 scenario. Provided that, the extra box should precipitate gypsum before reaching halite saturation. What are the periods required to reach gypsum and halite saturation points? How do they compare to the suggested insolation-paced alternations for marginal basins? How would the strait efficiency parameter impact these timescales? Unless I'm mistaken, such information does not exist in the present manuscript. Because the authors are aiming to relate their experiments with existing hypotheses, I suggest that these comparisons would be important. Is it possible to add a brief explanation of these to the manuscript?*

**We agree that including timescales in the manuscript would strengthen the manuscript. To achieve this, we added different time markers to the model output. Based on those results we can now analyze the timespans and durations for the different configurations and settings (see figure below, description not final)**

**To be able to interpret these figures correctly it is, however, important to highlight that the results describe a change that would result from a sudden change in restriction. A gradual change in restriction would lead to a different outcome. The influence of insolation is not included as time-varying forcing, instead we do explore the influence of a broad range of net-evaporation values on the steady state solution. It is, however, possible to infer their influence from previous studies that explored this effect (e.g. Ebner et al. 2024).**

**To elaborate on the question regarding the influence of strait efficiency we prepared preliminary versions of figures that could be included in the manuscript. They show the nonlinear behavior that is also expressed in the salinity plots (figure 2 in manuscript). Figure a describes the time the model takes to reach gypsum (solid line) or halite**

concentration (dashed line). The vertical asymptote of each curve intersects the x axis at the restriction parameter that would just not yet lead to gypsum or halite.  Figure b shows the timespan during which a model run would meet the conditions as defined in the manuscript. Here, the vertical asymptote of each curve marks those runs that meet the conditions once they have reached stability. i.e. the duration goes to infinity. To the left of this singularity, the model meets the conditions only for a short amount of time during the stabilizing phase.

[Figure]

*4. Technical corrections*

**We are focusing on comments that would benefit from a more elaborate answer**

*Technical corrections and minor comments: Title- A question of time and space: A model approach to the synchronicity of gypsum and halite deposition during the Messinian Salinity Crisis. Suggestion to add 'deposition' to the title.*

**We have decided to change the title to**
**Title- A question of time and space: A model approach to the synchronous precipitation of gypsum and halite deposits during the Messinian Salinity Crisis**

*7 – define Ma*

**When using it for the first time, we will define Ma as Million years before present.**

*Line 11 – …for different configurations… of what?*

**We plan to change the term configuration in the abstract through 'different precipitation patterns' and introduce the term in the introduction. This should prevent confusion and make the abstract easier to understand.**

*Line 20 – define kyr*

**We will use the non-shortened version of the duration and then translate to kyr when using it the first time.**

*Figure 1 – Add labels of different parameters (eg: evaporation, convection, diffusion) to one of the configurations. Strait of Gibraltar is not properly visible as you have not shown the Atlantic side.*

**We had decided against adding labels to the components to avoid cluttering of the figure. We will however add explanations of the symbols in the figure caption as well as elaborate on the meaning of the driver flux.**

**To make the location of the connection to the Atlantic clearer we can add an A for Atlantic and explain its meaning in the text. We want to avoid labeling the connection with 'strait of Gibraltar' as it is not clear where the connection to the Atlantic was located.**

*Model symbols and parameters – If the relative size of the extra box is f, shouldn't Aopen be (1-f)Atot and Aextra be (f)Atot?*

**Yes! We will correct that.**

*For Vextra, you have not prescribed what 500 m is (which, I assume is the depth/ thickness of the water column)*

**We will state in the text that our upper layer is 500m thick.**

*Line 135 – To arrange the diffusivity term in order → suggestion to rearrange the equation to $\square\,\square mix = Kmix\,.(Sopen - Sdeep)\,.Aopen\ dmix$ .*

**This is a good idea that will increase the readability of the equations.**

*Line 140 – Shouldn't the salt flux be upward, therefore for equations 5a and 5c the jmix terms become opposite in sign (positive for 5a and negative for 5c)?*

**This is indeed a translation error between the code and the manuscript. It is a sign error that will be corrected in the next version.**

*Lines 156 – should you mention the Black Sea as well?*

**We chose not to name the Black Sea here, since it is not clear whether the connection between those two was already established.**

*Figure 3c – Suggestion to update the label ca to ca2 (correct?)*

**The naming in this figure is indeed unfortunate. In this case we mean ca. as in circa. We will relabel this figure to make this more obvious.**

*Line 271 – can you state the salinity difference?*

**The maximum salinity difference varies greatly depending on the parameter values, as such we cannot give an absolute value, but we can give a range and elaborate on that.**

*Line 274 – You have not stated to which figure you are referring to.*

**We will refer again to figure 3b.**

*Line 362 – Should 'Sea' be removed?*

***Yes.***

*Line 383 – suggestion to add a figure explaining you time series of salinification.*

**We liked this idea a lot. See our answer to your question 2.**

---

## Author Comment (AC2)

**Dear reviewer,**

**First of all, we would like to thank you for your feedback, it will help us to strengthen our manuscript. Second, we would like to give a more detailed reaction to some of your comments. The answers will be highlighted in bold, while your questions are contrasted in cursive.**

*While the authors suggest that the model could be applied to other basins (e.g., the Red Sea), it is not clear how the specific model configurations (A1, A2, B) would translate to different geochemical settings. Could the model be adapted to explore other evaporite-forming basins more explicitly?*

**The model can be adapted by changing the dimensional parameters (area, depth), forcing (net evaporation) and restriction to fit the basin in question. Since our study focusses on the Mediterranean Sea, we chose values for the dimensional parameters to reflect that. The influence of other parameters (net-evaporation, restriction, relative size of the boxes) is tested by applying a range of possible values. We did not test the influence of dimensional parameters on the results since it was not within the scope of this research. However, those parameters can easily be changed and adapted by anyone who is interested in this aspect.**

*Regarding the title, you should mention the "Mediterranean Sea" because the MSC occurred in the Med Sea, and your study focused on the Med Sea.*

**We chose not to mention the Mediterranean Sea explicitly, since the title is already on the verge of being too bulky. We do not think adding this information would increase the information density of the title as the term 'Messinian Salinity Crisis' is indeed already strongly connected to the Mediterranean Sea,**

*The paper mentions that constant evaporation rates were used. How might a variable evaporation rate, could impact the model results? Could this change the timing or locations of gypsum and halite precipitation? Were there any sensitivity tests performed to explore this?*

**This is a very good point. As reaction to this and the other reviews we are going to expand our analysis by elaborating more on the time component.**

**To approach this, we add the times it would take to reach gypsum and halite precipitation respectively, as well as the time a model run stays within conditions that describe simultaneous but locally separated precipitation of Gypsum and Halite. The visualization of this dataset will be added to the set of plots in Figure 2 (see below).**

[Figure]

**We do not explore the influence of a variable evaporation rate because this would increase the amount of unknows in our study.**

*The manuscript does not provide sufficient discussion on the role of the Strait of Gibraltar in influencing Mediterranean circulation and salinity. A more detailed analysis of how restricted or variable water exchange through the Strait affects gypsum and halite precipitation patterns would add depth to the study.*

**We avoided labeling the connection to the Atlantic as Strait of Gibraltar, as the exact location of the connection between the Mediterranean Sea and the Atlantic is not entirely clear, with the Betic and Riffean corridor being two likely candidates.**

**We do however explore the influence of the efficiency of the connection independent of its location. This is expressed in the descriptor $R_q$ as well as the strait efficiency. We will add a sentence to highlight this.**

**In connection with the analysis of the time component we will also add one more point to our conclusion. Since the process of restriction over time has a non-linear influence on the rate of change of the salinity in the different boxes, it is crucial to understand the closure of the connection better. Without having a better grasp on this process, the number of possible scenarios is unlimited. This however does not change the main conclusion of this study. Simultaneous precipitation of gypsum in the periphery and halite in the deep basin is only possible in a basin close to halite saturation.**

*Have you conducted sensitivity tests on key parameters such as evaporation rates, river water composition, or Strait of Gibraltar exchange? If not, how might these factors impact your results?*

**We do not label it as such, but we do explore the influence of the interplay of restriction and evaporation rates (discussed in 206- 237,  272-2076, ), as well as the influence of other parameters (276 -281, 288 – 301, 303-308).**

**The analysis of the influence of river water composition on scenario B+ will be supported by a table for the exact values and a more in depth description on how those values relate to the results.**

*The abstract could benefit from a clearer articulation of the novelty of the study. It touches on known issues but doesn't strongly emphasize how the modeling results diverge from or contribute to existing theories.*

**We will strengthen the message by highlighting the time component and the need for a better understanding of the change in restriction.**

*The comparison with Simon & Meijer (2017) is helpful, but the contributions of the present study (e.g., density driven dynamic overturning) could be more explicitly emphasized early on. For instance, the detailed breakdown of different studies (e.g., Meilijson et al., 2019 vs. Manzi et al., 2018) could be summarized more concisely to avoid overloading the reader with too many specific comparisons at the outset.*

*Citations are included in parentheses, but in some cases, they interrupt the flow of the text. For better readability, consider rephrasing sentences to integrate citations more naturally. Example: Instead of "5.97 to 5.33 Ma, (Roveri et al., 2008)," you could say "According to Roveri et al. (2008), the event occurred between 5.97 and 5.33 Ma." This would make the text*

*smoother. Consistency in citation formatting is needed. For example, in some instances, authors' names are written in all caps, which should be corrected., e.g. (Decima & WEZEL, 1971; Decima & Wezel, 1973) .*

*The flow between ideas could be improved with clearer transitions between sections. For example, when moving from the discussion of modeling to the thermo-haline circulation section, adding transitional sentences can help guide the reader more smoothly from the background after the modeling approach.*

**Those are good ideas; we agree that this will increase readability.**

*The conversion from Atlantic water to more saline Mediterranean overflow water happens via an overturning cell in the Mediterranean Sea." Not clear, this sentence could be rephrased.*

**The process is elaborated on in the next sentence. We will rephrase the sentence to**

**'The conversion from Atlantic water to more saline and warmer Mediterranean overflow water (MOW) can be described via an overturning cell in the Mediterranean Sea.'**

*The abbreviation "MSC" for Messinian Salinity Crisis is introduced but not consistently used throughout the text. It would help to use the abbreviation after it's introduced to avoid repeating the full term, e.g. line 342.*

**We agree with this.**

*Method section*

*The overall structure and technical content are strong, but enhancing transitions will improve readability. While you define many variables, key terms could be better explained to ensure the reader fully understands. For example, explaining "net evaporation rate" in more detail would help if a reader is not familiar with the exact context. Similarly, more context around κ and why it's used differently from its traditional sense could be provided upfront to avoid confusion.*

*Some terms such as "anti-estuarine circulation," "driver flux," and "marginal basin" are used without sufficient context for non-expert readers.  After describing each configuration (A1, A2, and B), it might be helpful to summarize their key differences in a table. This would help the reader quickly differentiate between them.*

**The concept of the driver flux will now also be introduced in the description of Figure.**

**Anti-estuarine circulation can be introduced with additional information 'i.e. outflow more saline than inflow' in line 62.**

*What is the temporal resolution of your model, and how does influence the results, particularly regarding the timing of halite and gypsum precipitation?*

**The model itself operates with dt = 0.5years, we also tested different timesteps   up to  2 and saw no changes in the results.**
**To address the issue of timing we have now calculated the time the model takes to reach halite and gypsum, as well as the timespan the conditions for simultaneous precipitation are met. We plan to include the figures in the results section (Figure 2) and their implications in the discussion.**

**Figure a describes the time the model takes to reach gypsum (solid line) or halite concentration (dashed line). The vertical asymptote of each curve intersects the x axis at the restriction parameter that would just not yet lead to gypsum or halite.  Figure b shows the timespan during which a model run would meet the conditions as defined in the manuscript. This time the vertical asymptote of each curve marks those runs that meet the conditions once they have reached stability. i.e. the duration goes to infinity. Left from this singularity, the model meets the conditions only for a short amount of time during the stabilizing phase.**

[Figure]

*Results section*

*The use of the strait restriction parameter ($q$) and its bulky unit [($m^3/s$)/($\sqrt{kg/m^3}$)] is well justified, but simplifying its interpretation would help make the section more accessible.*

**We are omitting the bulky unit later on to not overload the reader, but we can also add a brief explanation of the unit itself (i.e. relating water flux m³/s to the square root of salinity difference sqrt(kg/m³ ) ).**

*The model uses generic assumptions about river water composition to assess gypsum precipitation in the extra box. How significant are variations in river chemistry (e.g., calcium and sulfate concentrations) for altering the results, and were sensitivity tests performed with different river compositions?*

**To make their influence clearer we will add a table showing the compositions we used.**

**This table will then be used to guide a more elaborate analysis of the results. The main message will be that net evaporation can be in compared to the river inflow before halite starts precipitating, the higher the calcium and sulphate concentrations are in comparison to natrium and chloride.**

*The section compares the model results with the Mediterranean and Red Seas, I think that the appearance of the part about the Black Sea is very abrupt, and there is very little information about the Black Sea in the paper.*

*Discussion section*

*The discussion is rich in technical detail but sometimes lacks a clear "so what?" moment that emphasizes why these results are significant in the context of the Messinian Salinity Crisis or other studies on evaporite formation. While the model's limitations are well discussed, it*

*would be helpful to suggest what future studies could address based on these results. How could the model be improved? What future work is needed to fill the gaps identified in your study?*

*Conclusion section*

*The conclusion, while summarizing the key findings, could be strengthened by tying the results more explicitly to potential future research directions or practical implications. It currently ends somewhat abruptly and could benefit from a more definitive closing statement on the significance of the study.*

**In connection with the more elaborate treatment of the time component we will highlight the necessity to describe the process of restriction over time to be able to describe the problem in more detail. We will also put more emphasis on our conclusion that simultaneous precipitation of halite and gypsum most likely only occurred towards the end of stage 1 of the consensus model and not sooner.**

*For example, what does this timeline and model tell us about the general understanding of evaporite formation in restricted basins? How might these findings inform future models or field studies in similar settings?*

**We will add that our timeline is an addition to the bullseye patten that is observed in deposits formed by a drying lake.**

*Legend of figures in bold?*

*Ensure that table legends appear at the top of the tables. This would align the manuscript with common publication standards.*

**In both instances I had misinterpreted the template. This will be fixed in the next version.**

*A few suggestions:*

**The comments listed under this header did not need clarification and will not be discussed here. They will be taken into account when revising the manuscript.**

---

## Author Comment (AC3)

**Dear reviewer 3,**

**We would like to thank you for your thoughtful comments. We have discussed them extensively and reply to them in detail below.**

*1. The box model construction illustrated in Figure 1 shows two-way Mediterranean Atlantic exchange. It is widely accepted that this configuration probably only applies to Stage 1 of the MSC, when gypsum was precipitated in the marginal basins of the Mediterranean requiring a high sea-level. Stage 2 and 3 are more likely to have occurred with Atlantic inflow but negligible outflow from the Med, consistent with a base level that was below the level of the gateway. Consequently, the main application of this model configuration is Stage 1. This is mentioned in the abstract but is not made clear in the introduction where a description of all three phases of the MSC (L31-39) is followed by a statement about the challenges of shallow-deep water correlation as a justification for looking at synchronous gypsum-halite precipitation (L50-50).*

**This is correct. The model only applies to a Mediterranean that is still connected to the Atlantic Ocean. Our analysis thus focuses on what is referred to as phase 1 in the consensus model and is commonly attributed with only gypsum precipitation. This assumption is often challenged.**

**Place our study within we will add the phase we are focusing on in lines 40 and following as well as an additional figure to visualize the difference between consensus model and alternative scenario.**

[Figure]

| **Alternative scenario of synchronous precipitation of gypsum and halite** | **Consensus model** |
| --- | --- |

*2. The paper concludes that synchronous precipitation of gypsum and halite can only happen in Scenario A when the system as a whole is close to halite saturation. While I accept the statement in the first paragraph of the discussion (L334-339) that the model is not meant to represent "the complexity of the Mediterranean Sea", none the less, it is possible at least to point out the episodes within the MSC that are closet to the model configuration used and consider the implications. For example, some discussion about when within Stage 1 reaching near halite saturation is most likely would enhance the applicability of the results. The strait efficiency required to generate synchronous gypsum-halite precipitation could be evaluated*

*against the Sr isotope ratio data for Stage 1 which progressively diverges from the ocean water curve. This might then enable them to evaluate the duration of the potential overlap between Stage 1 and 2 mentioned in L417-20.*

**Comparing model results to strontium values is indeed an interesting idea, that we are addressing by developing a model that accounts for spatial differences (east vs. west, central vs. marginal), the influence of a non-constant freshwater balance (FWB), river chemistry, and the role of the Paratethys in the evolution of strontium isotopic ratios and the formation of evaporite deposits. Without a more detailed analysis of these factors, any comparison to strontium values appears futile, as their average remains relatively stable throughout stage 1.**

**In our discussion we already state that conditions of simultaneous precipitation likely occurred toward the end of stage 1 of the consensus model. To elaborate on this our analysis will include new aspects, such as the timespans required for the model to reach gypsum and halite saturation, as well as the duration for which synchronous precipitation of gypsum and halite conditions are maintained during each model run. Those new results (see below) will be added to the current Figure 2.**

**Figure a describes the time the model takes to reach gypsum (solid line) or halite concentration (dashed line). The vertical asymptote of each curve intersects the x axis at the restriction parameter that would just not yet lead to gypsum or halite. Figure b shows the timespan during which a model run would meet the conditions as defined in the manuscript. This time the vertical asymptote of each curve marks those runs that meet the conditions once they have reached stability. i.e. the duration goes to infinity. Left from this singularity, the model meets the conditions only for a short amount of time during the stabilizing phase.**

[Figure]

*3. Section 3.2.3 (Scenario B) – this section needs a little more explanation of the chemistry and particularly some more information about the chemistry of the rivers that are modelled in Fig. 4 so that the reader can see how their different compositions result in different consequences.*

We will add a reference to the table with the raw values in Gaillardet et al. (1999) and expand on how this information is used within our simplified description of salinity.

To not only make our approach easier to follow but also add depth to the results, we will add a brief discussion on how those compositions influence the results.

| name | NaCl | CaSO$_4$ |
|---|---|---|
| No ions | 0 | 0 |
| Rhone | 0.03 | 0.07 |
| Po | 0.03 | 0.09 |
| Nile | 0.07 | 0.07 |
| Ebro | 0.12 | 0.19 |
| Mediterranean Sea | 271.1 | 5.25 |

*technical issues*

*Title – suggested tweak "A model approach to the synchronous precipitation of gypsum…."*

We have decided to change the title to
Title- A question of time and space: A model approach to the synchronous precipitation of gypsum and halite deposits during the Messinian Salinity Crisis

*Abstract L9 – "different configurations". A little more clarity about what those configurations are would help here*
We will clarify that we talk about precipitation patterns in the abstract and elaborate on this in the introduction.

*Figure 3 – both the caption and the text in line 244 state that there should be an "x" showing the present-day Mediterranean on Fig 3a, but I can't see it.*
It is indeed very small (almost on the x axis, right above 70 cm/yr), we will make it more obvious.

*Section 4.2 L405 – the thickness of the lower Tripoli Unit in the Lorca basin is used to illustrate the likely sedimentation rates resulting from step 3, but that isn't helpful if you don't know how thick that unit is…. and I don't!*

*This is a very good point. We will rewrite this argument by adding this information.*

*The interval in question is 5m thick and has been attributed to an interval of 400kyr. The resulting precipitation rate, however, is questioned since the base of this sediment cannot be defined due to a gap in sedimentation (Rouchy et al., 1998).*

---

## Author Response (AR1)

**Author's answers to comments of the first revision and changes applied to the new version of the manuscript**

**Most notable changes made**

We added the analysis of the time component of our model to the results& analysis as well as the discussion and the conclusion, with this we address comments made by all 3 referees.

**Answers to extended comments**

**Rev 1**

1. A proper comparison with the real-world scenario
   The aim of the authors is to test if their simplified model predicts whether the salt units (gypsum and halite) in marginal and deep basins could have been precipitated concurrently. Even though they briefly explain the existing hypotheses for the depositional patterns within the MSC, I suggest that proper definitions of marginal versus deep basins should be provided prior to elaborating their experiments, with suitable diagrams of existing models for depositional units in each setting (for example, a simplified version of Roveri et al (2014) synthesis). Such insight would make it easier for the reader to follow the author's intentions, in relation to their model experiments. I propose that the same diagram may be used to show the ambiguity in horizontal continuity of the PLG, as the authors indicate in their introduction.
   **We have added a new figure to visualize the differences between the two conceptual models we are exploring and have added a clarification in the introduction.**

2. Organization
   Added to my above suggestion, the manuscript may be ordered in the following sequence: A general introduction to the MSC, with explanations on marginal versus deep basins; Existing hypotheses for the different timing of sedimentary unit deposition (including diagrams); Methods; Results; Discussion – here, I suggest including a better explanation of their model results with respect to actual observations they provided in the revised introduction. Under implications, the authors combine their different configurations to develop a timeline of salinification. I suggest adding a new diagram to explain their timeline, as this is one of their final interpretations, and Conclusions.
   **We have added a figure to visualize the timeline. Our discussion of the time component also strengthens our message now.**

3. Timescales
    A majority of MSC researchers suggest that the PLG unit was developed as an alternating sequence of gypsum-marl couplets, with up to 17 units paced by insolation (Lugli et al., 2010,

Manzi et al., 2013). For someone who may try to compare the suggested scenarios with existing timescales of PLG/ Halite unit deposition (eg: PLG stage during 5.97-5.60 Myr, Halite deposition during 5.60 5.55 Myr), no information has been provided regarding the timescales considered for model experiments. For instance, it has not been shown how long it will take to reach halite saturation in the extra box in A1 scenario. Provided that, the extra box should precipitate gypsum before reaching halite saturation. What are the periods required to reach gypsum and halite saturation points? How do they compare to the suggested insolation-paced alternations for marginal basins? How would the strait efficiency parameter impact these timescales? Unless I'm mistaken, such information does not exist in the present manuscript. Because the authors are aiming to relate their experiments with existing hypotheses, I suggest that these comparisons would be important. Is it possible to add a brief explanation of these to the manuscript?

**We have added a detailed description and discussion of the time component to address this.**

**Rev 2**

1. While the authors suggest that the model could be applied to other basins (e.g., the Red Sea), it is not clear how the specific model configurations (A1, A2, B) would translate to different geochemical settings. Could the model be adapted to explore other evaporite-forming basins more explicitly?

   **Since our focus is on the Mediterranean Sea and the MSC we have decided to not add an extra example.**

2. Regarding the title, you should mention the "Mediterranean Sea" because the MSC occurred in the Med Sea, and your study focused on the Med Sea.

   **We chose not to mention the Mediterranean Sea explicitly, since the title is already on the verge of being too bulky. We do not think adding this information would increase the information density of the title as the term 'Messinian Salinity Crisis' is indeed already strongly connected to the Mediterranean Sea.**

3. The paper mentions that constant evaporation rates were used. How might a variable evaporation rate, could impact the model results? Could this change the timing or locations of gypsum and halite precipitation? Were there any sensitivity tests performed to explore this?

   **We now refer to another paper that explored the influence of a varying net evaporation rate**

4. The manuscript does not provide sufficient discussion on the role of the Strait of Gibraltar in influencing Mediterranean circulation and salinity. A more detailed analysis of how restricted or variable water exchange through the Strait affects gypsum and halite precipitation patterns would add depth to the study.

   **We avoided labelling the connection to the Atlantic as Strait of Gibraltar, as the exact location of the connection between the Mediterranean Sea and the Atlantic is not entirely clear, with the Betic and Riffean corridor being two likely candidates.**
   **The influence of restricted exchange is already explore in the manuscript and we address the influence of a changing restriction (L450)**

5. Have you conducted sensitivity tests on key parameters such as evaporation rates, river water composition, or Strait of Gibraltar exchange? If not, how might these factors impact your

results?

**We have added the compositions of the rivers we used and extended the description of the results. The influence of the other parameters was already explored in the manuscript**

6. Missing punctuation occurs in multiple sentences where commas or periods could help separate clauses or clarify meaning (references style, the caption of the figure in bold, the table legends …).

**Several changes have been made to improve this.**

7. The abstract could benefit from a clearer articulation of the novelty of the study. It touches on known issues but doesn't strongly emphasize how the modeling results diverge from or contribute to existing theories.

**We improved the way the methods are introduced in the abstract to highlight the novelty of our approach.**

8. The comparison with Simon & Meijer (2017) is helpful, but the contributions of the present study (e.g., density driven dynamic overturning) could be more explicitly emphasized early on. For instance, the detailed breakdown of different studies (e.g., Meilijson et al., 2019 vs. Manzi et al., 2018) could be summarized more concisely to avoid overloading the reader with too many specific comparisons at the outset.

**We have added an extra figure to the introduction to visualise the premise of out research question.**

9. Citations are included in parentheses, but in some cases, they interrupt the flow of the text. For better readability, consider rephrasing sentences to integrate citations more naturally. Example: Instead of "5.97 to 5.33 Ma, (Roveri et al., 2008)," you could say "According to Roveri et al. (2008), the event occurred between 5.97 and 5.33 Ma." This would make the text smoother.

**We have rephrased that sentence.**

10. Consistency in citation formatting is needed. For example, in some instances, authors' names are written in all caps, which should be corrected., e.g. (Decima & WEZEL, 1971; Decima & Wezel, 1973)

**We have fixed this problem**

11. The flow between ideas could be improved with clearer transitions between sections. For example, when moving from the discussion of modeling to the thermo-haline circulation section, adding transitional sentences can help guide the reader more smoothly from the background after the modeling approach.

**we have not added transitional sentences**

12. The conversion from Atlantic water to more saline Mediterranean overflow water happens via an overturning cell in the Mediterranean Sea." Not clear, this sentence could be rephrased.

**We have rephrased the sentence.**

13. The abbreviation "MSC" for Messinian Salinity Crisis is introduced but not consistently used throughout the text. It would help to use the abbreviation after it's introduced to avoid repeating the full term, e.g. line 342.

**We are now only using MSC after it has been introduced.**

14. While you define many variables, key terms could be better explained to ensure the reader fully understands. For example, explaining "net evaporation rate" in more detail would help if a reader is not familiar with the exact context. Similarly, more context around κ and why it's used differently from its traditional sense could be provided upfront to avoid confusion. Some

terms such as "anti-estuarine circulation," "driver flux," and "marginal basin" are used without sufficient context for non-expert readers.

**net evaporation rate: we chose to not introduce that term as we assume that our readers are familiar with that term.**

**Anti-estuarine circulation is now introduced**

**driver flux is now introduced with the model sketch**

15. After describing each configuration (A1, A2, and B), it might be helpful to summarize their key differences in a table. This would help the reader quickly differentiate between them.

**We found it to be not beneficial to add a table for that and trust that the introduction and sketches are sufficient.**

16. What is the temporal resolution of your model, and how does influence the results, particularly regarding the timing of halite and gypsum precipitation?

**The temporal resolution is given in the table displaying the parameters and has no influence on the timing. However we have added a detailed discussion of the timing of precipitation.**

17. The use of the strait restriction parameter ($q$) and its bulky unit [$(m^3/s)/(\sqrt{kg/m^3})$] is well justified, but simplifying its interpretation would help make the section more accessible.

**We did not know how to simplify this.**

18. The model uses generic assumptions about river water composition to assess gypsum precipitation in the extra box. How significant are variations in river chemistry (e.g., calcium and sulfate concentrations) for altering the results, and were sensitivity tests performed with different river compositions?

**We have added an overview of the compositions used, as well as a description of their influence on the outcome.**

19. The section compares the model results with the Mediterranean and Red Seas, I think that the appearance of the part about the Black Sea is very abrupt, and there is very little information about the Black Sea in the paper.

**The Red Sea is just to add another example, next to present day Mediterranean Sea, to help the reader get a feeling for the scale of the metric we are using and to show how extreme the restriction during the MSC must have been.**

20. The discussion is rich in technical detail but sometimes lacks a clear "so what?" moment that emphasizes why these results are significant in the context of the Messinian Salinity Crisis or other studies on evaporite formation.

**We hope to have strengthened that point by adding the discussion of the time component.**

21. It would be helpful to suggest what future studies could address based on these results. How could the model be improved? What future work is needed to fill the gaps identified in your study?

**We now address this in the conclusion.**

22. The conclusion, while summarizing the key findings, could be strengthened by tying the results more explicitly to potential future research directions or practical implications. It currently ends somewhat abruptly and could benefit from a more definitive closing statement on the significance of the study. For example, what does this timeline and model tell us about the general understanding of evaporite formation in restricted basins? How might these findings inform future models or field studies in similar settings?

**We have extended our conclusion to address this comment.**

**Rev 3**

1. The box model construction illustrated in Figure 1 shows two-way Mediterranean Atlantic exchange. It is widely accepted that this configuration probably only applies to Stage 1 of the MSC, when gypsum was precipitated in the marginal basins of the Mediterranean requiring a high sea-level. Stage 2 and 3 are more likely to have occurred with Atlantic inflow but negligible outflow from the Med, consistent with a base level that was below the level of the gateway. Consequently, the main application of this model configuration is Stage 1. This is mentioned in the abstract but is not made clear in the introduction where a description of all three phases of the MSC (L31-39) is followed by a statement about the challenges of shallow-deep water correlation as a justification for looking at synchronous gypsum-halite precipitation (L50-50).

   **We have added a figure to make this clearer.**

2. The paper concludes that synchronous precipitation of gypsum and halite can only happen in Scenario A when the system as a whole is close to halite saturation. While I accept the statement in the first paragraph of the discussion (L334-339) that the model is not meant to represent "the complexity of the Mediterranean Sea", none the less, it is possible at least to point out the episodes within the MSC that are closet to the model configuration used and consider the implications. For example, some discussion about when within Stage 1 reaching near halite saturation is most likely would enhance the applicability of the results. The strait efficiency required to generate synchronous gypsum-halite precipitation could be evaluated against the Sr isotope ratio data for Stage 1 which progressively diverges from the ocean water curve. This might then enable them to evaluate the duration of the potential overlap between Stage 1 and 2 mentioned in L417-20.

   **Comparing model results to strontium values is indeed an interesting idea, that we are addressing by developing a model that accounts for spatial differences (east vs. west, central vs. marginal), the influence of a non-constant freshwater balance (FWB), river chemistry, and the role of the Paratethys in the evolution of strontium isotopic ratios and the formation of evaporite deposits. Without a more detailed analysis of these factors, any comparison to strontium values appears futile, as their average remains relatively stable throughout stage 1.**

   **To give a better indication of the timespan we have added the times we can deduct from the model to the results and discussion.**

3. Section 3.2.3 (Scenario B) – this section needs a little more explanation of the chemistry and particularly some more information about the chemistry of the rivers that are modelled in Fig. 4 so that the reader can see how their different compositions result in different consequences.

   **We have added the compositions and extended the explanation of the results to address this.**

**Comments by line**

| location | comment | answer |
|----------|---------|--------|
| title | suggested tweak "A model approach to the synchronous precipitation of gypsum…." | |

| | | |
|---|---|---|
| | A question of time and space: A model approach to the synchronicity of gypsum and halite deposition during the Messinian Salinity Crisis. Suggestion to add 'deposition' to the title. | We have changed out title to *A question of time and space: A model approach to the synchronous precipitation of gypsum and halite during the Messinian Salinity Crisis* |
| L4 | Earth Sciences – 2 words | corrected |
| L6 | well studied…. | Corrected |
| | "Saltgiants" => "Salt giants" ? | corrected |
| L7 | Define Ma | Removed from line 7 and added to line 31 |
| L9 | "different configurations". A little more clarity about what those configurations are would help here | Changed to: different possible configurations of the basin and circulation |
| | "could be not yet been confirmed" => "could not yet be confirmed". | corrected |
| L11 | …for different configurations… of what? | Now specified |
| L16 | "salinifying". Suggested alternative "a timeline for an increasingly saline basin." | We chose to keep this term throughout the manuscript |
| L17 | …a sufficiently restricted marginal basin…. | Changed throughout the text |
| L18 | remove 'the one of' and 'areas of the' → gives- …once the average salinity approaches halite saturation it can also form in the open basin… | changed |
| L19 | same as for Line 17 | Changed throughout the text |
| L20 | Define kyr | Defined in line 31 |
| L21 | …within a one basin… rephrase | corrected |
| L27 | change to …youngest salt giant formation… | Changed |
| L28-29 | suggestion to reorganize for clarity | Changed |
| L32 | "reaches up to three km" => "reaches up to three kilometers". | Changed |

| L40 | "unambiguous, since we, for example, cannot follow" should be "unambiguous since, for example, we cannot follow". | changed |
|---|---|---|
| L47-48 | vague statement …a more recent study, however, reopened this question again… | Extended with content of study |
| L48 | "re-opened" => "reopened". | changed |
| L56 | should you mention the Black Sea as well? | In this line we are referring to a paper that focused on the Mediterranean Sea, while there are box model studies that focus on the Black Sea, none of them would be as good of a comparison as (Simon & Meijer, 2017) |
| L80 | "overturing" => "overturning". | corrected |
| L95 | add citations: From previous studies, we know… | Added 3 studies |
| L135 | To arrange the diffusivity term in order → suggestion to rearrange the equation to $\mathord{?}\,?mix = Kmix\,.(Sopen - Sdeep).Aopen\ dmix$ | changed |
| L140 | Shouldn't the salt flux be upward, therefore for equations 5a and 5c the jmix terms become opposite in sign (positive for 5a and negative for 5c)? | changed |
| L271 | can you state the salinity difference? | added |
| L274 | You have not stated to which figure you are referring to. | Added the reference |
| L283 | perhaps move 'also' in front of 'halite'? | We decided against that, because it would shift the meaning of the sentence |
| L300 | Reads disorganized when you start the sentence with 'which'. | changed |
| L334 | "The models presented here not a representation of the complexity..." => "The models presented here are not a representation..." | corrected |
| L340 | a space between "per" and "1°C" (per 1°C) | corrected |

| L362 | the word "Dead" is missing before "Sea" | Removed 'Sea' |
|---|---|---|
| | Should 'Sea' be removed? | |
| | Reads disorganized when you start the sentence with 'those'. | rephrased |
| L363 | …to be represented? | changed |
| L383 | suggestion to add a figure explaining you time series of salinification. | Added a figure in line 422 |
| L388 | Into the deep basin? | Changed to 'deep basin' |
| L391 | Into the deep basin? | |
| L405 | – the thickness of the lower Tripoli Unit in the Lorca basin is used to illustrate the likely sedimentation rates resulting from step 3, but that isn't helpful if you don't know how thick that unit is…. and I don't! | Information is now added |
| L427 | Majorly? | corrected |
| 428 | Explain why your results do not exclude this (vague statement). | We made this clearer by adding more information |

**Comments regarding figures and tables**

| Figure 1 | Add labels of different parameters (eg: evaporation, convection, diffusion) to one of the configurations. Strait of Gibraltar is not properly visible as you have not shown the Atlantic side. | We have added the label 'Atlantic' to indicate the position of the connection. We have decided against labelling the parameters within the sketches as they would otherwise be too convoluted |
|---|---|---|
| Figure 2d | for clarity, label the y-axis "salinity difference" | |
| Figure 3 | both the caption and the text in line 244 state that there should be an "x" showing the present-day Mediterranean on Fig 3a, but I can't see it. | The x was indeed hard to spot, we have replaced it by a bigger, red x |
| Figure 3c | Suggestion to update the label ca to ca2 (correct?) | The labelling was confusing, we have corrected that. |
| Table 1 | If the relative size of the extra box is f, shouldn't Aopen be (1-f)Atot and Aextra be (f)Atot? | We have corrected that. |
| | For Vextra, you have not prescribed what 500 m is (which, I assume is the depth/ thickness of the water column) | We have added this information now to the text to make this clearer |

---

## Author Response (AR2)

Dear Dr. Völker,

we would like to thank you for your helpful comments. We have corrected the problems you pointed out and included a reference to the paper you suggested.

Kind regards,

Ronja Ebner